# HIERARCHICAL POLICY LEARNING FOR LONG HORIZON INTERACTIVE INSTRUCTION FOLLOWING

## ABSTRACT

Robotic agents performing domestic chores using natural language directives require to learn a complex task of navigating an environment and interacting with objects in it. To address such composite tasks, we propose a hierarchical modular approach to learn agents that navigate and manipulate objects in a divide-and-conquer manner for the diverse nature of the entailing tasks. Specifically, our policy operates at three levels of hierarchy. We first infer a sequence of subgoals to be executed based on language instructions by high-level *policy composition controller* (PCC). We then discriminatively control the agent's navigation by a *master policy* by alternating between navigation policy and various independent interaction policies. Finally, we infer manipulation actions with the corresponding object masks using the appropriate *interaction policy*. Our hierarchical agent, named *HACR (Hierarchical Approach for Compositional Reasoning)*, generates a human interpretable and short sequence of sub-objectives, leading to efficient interaction with an environment, and achieves the state-of-the-art performance on the challenging ALFRED benchmark.

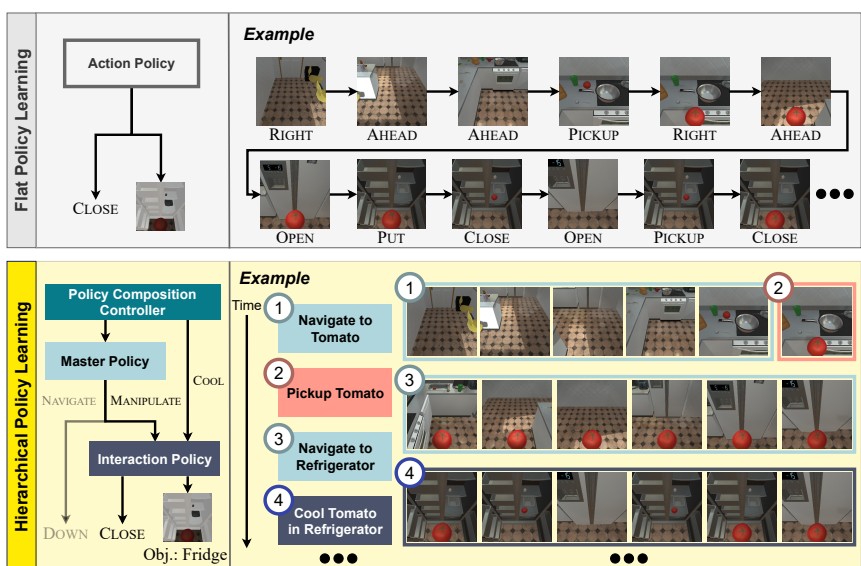

Figure 1: 'Flat policy learning' vs. proposed 'Hierarchical policy learning'. The flat policy learning has been employed in prior arts, training an agent to directly learn the low-level actions. In contrast, our hierarchical policy decomposes a long-horizon task into multiple subtasks and leverages the high-level abstract planning, which enables an agent to better address long-horizon planning.

# 1 INTRODUCTION

For the long-awaited dream of building a robot to assist humans in daily life, we now witness rapid advances in various embodied AI tasks such as visual navigation (Anderson et al., 2018; Chen et al.,

2019; Chaplot et al., 2017; Krantz et al., 2020), object interaction (Zhu et al., 2017; Misra et al., 2017) and interactive reasoning (Das et al., 2018; Gordon et al., 2018). Towards building such realistic robotic assistant, the agent should be capable of all of these tasks to address more complex problems. A typical approach for combining these abilities is to build a unified model (Shridhar et al., 2020; Singh et al., 2021) to jointly perform different sub-tasks. For example, the type of reasoning necessary for navigation differs significantly from the reasoning required for object interaction; the former focuses on inferring the free space and a search strategy to reach a target location while the latter requires detection of objects and inferring their distances and states (Singh et al., 2021). Meanwhile, human cognition process learns to divide a task into sub-objectives such as navigation or interaction, which enables humans to facilitate complex reasoning in a variety of circumstances (Hayes-Roth & Hayes-Roth, 1979). Inspired by those, we propose a hierarchical agent that learns and infers low-level policies for different sub-tasks. To evaluate our proposal in challenging scenarios, we particularly focus on the task of interactive instruction following, where the goal is to execute a set of navigation and object interaction tasks specified by natural language. This task poses numerous challenges including long-term planning, partial observability, and irreversible state changes. To complete a task successfully, an agent needs to navigate through hundreds of action sequences and interact with various objects in novel environments (*e.g.*, dining room, and kitchen *etc.*) which necessitates thorough comprehension of both visual scenes and natural language instructions.

Specifically, we propose to disentangle the task into high-level subgoals which are executed by the respective subgoal policies. Our agent is composed of (1) a policy composition controller (PCC) that specifies sub-policy execution sequence, (2) a master policy (MP) that specialises in navigation and provides a signal to conduct interaction tasks, and (3) several interaction policies (IP), which execute autonomous low-level interactions for each subgoal type. This disentanglement results in easier to analyze subtasks with shorter horizons.

Inspired by visual navigation literature (Yang et al., 2018; Wortsman et al., 2019) that encodes target object information for navigation, we propose an *object encoding module* (OEM) that provides such target object information and serves as a navigation subgoal monitor. During navigation to locate an object, the agent faces uncertain obstacles that forces it to traverse in a loop (deadlock state). We additionally propose a loop escape module (LEM) that assist the agent in circumventing such obstacles, based on a stored history of prior visual observations, thus resulting in better navigation.

We summarize our contributions as follows;

- We propose a hierarchical framework that decomposes the instruction following task into semantic subgoals and effectively addresses them with corresponding submodules.

- We propose object encoding module (OEM) that encodes object information from natural language instructions for effective navigation and a loop escape module (LEM) to escape a deadlock state with a stochastic policy.

- The proposed agent achieves state-of-the-art performance on the ALFRED benchmark.

## 2    RELATED WORK

Prior works propose numerous task setups and benchmarks for developing an agent to complete complicated tasks given natural language directives, such as agents trained to navigate (Li et al., 2020; Xu et al., 2021) or solve household tasks (Shridhar et al., 2020). However, the vast majority of approaches for these tasks employ a flat reasoning (Singh et al., 2021; Nguyen et al., 2021), in which the agent decides on the low-level actions accessible while moving through the environment (Gupta et al., 2017; Zhu et al., 2020). When the prior arts seek to define subtasks, they are frequently defined in a shallow manner with only two layers of hierarchy (Zhang & Chai, 2021; Corona et al., 2020; Blukis et al., 2021; Andreas et al., 2017; Gordon et al., 2018; Yu et al., 2019; Das et al., 2019). Furthermore, these strategies are inefficient in terms of data requirements. Natural Language is subjective, and even a seemingly simple command can contain several unstated meanings. Because of this semantic gap, most present solutions involve either a large amount of labelled data or trial-and-error learning to map language to low-level actions. Compounding errors also impede performance for activities with a long and complex horizon (Xu et al., 2019). We introduce a system for representing and using hierarchical knowledge in order to better the control of embodied agents doing complex tasks stated in natural language directives. Furthermore, because

of the modular structure, our agent reasons and accomplishes tasks along longer paths than previous studies, spanning numerous subgoals.

The task of interactive instruction following necessitates not only navigation but also the interaction with surrounding objects by following natural language directives. (Shridhar et al., 2020) proposed a CNN-LSTM-based baseline agent with progress tracking (Ma et al., 2019). (Singh et al., 2021) offer a modular strategy that factorises perception for mask prediction and policy for action prediction. (Nottingham et al., 2021) offer a modular system in which language encoding, visual state extraction, and action prediction are all trained independently in different modules. (Nguyen et al., 2021) suggested a system for predicting actions and masks corresponding to different instructions independently. They combine numerous visual inputs from navigable directions to provide a larger field of view for improved navigation. (Zhang & Chai, 2021) propose a Transformer-based hierarchical agent whereas (Suglia et al., 2021) present a Transformer-based agent that uses object landmarks for navigation. (Pashevich et al., 2021) present a Transformer-based agent that uses a multimodal Transformer for exploiting the multiple input modalities. (Blukis et al., 2021) propose a technique for successful high-level reasoning by reconstructing a 3D map for encode spatial semantic representation. Finally, a modular policy with a two levels of hierarchy for instruction following have previously been proposed by (Corona et al., 2020) which results in poor performance on long-horizon task. In contrast, our policy operates at three hierarchical levels exploiting the fact that navigation and interaction are semantically diverse activities that requires independent processing.

## 3 APPROACH

We observe that the visual information for navigation varies considerably over time while for object interaction it is largely stationary. As a consequence, the agent benefits from learning different policy modules for the two tasks, because the former needs to reason about temporal history and global environment information, whilst the latter requires to focus on local visual cues for mask prediction. Furthermore, the class imbalance between navigation and interaction atomic actions (*i.e.*, navigation actions are far more frequent than the interaction actions) would bias a model towards more frequent navigation actions.

Inspired by these motivations, we design our modular architecture with three levels of hierarchy; (1) A high-level policy composition controller (PCC) that uses language instructions to generate a sequence of sub-objectives, (2) a master policy that specialises in navigation and determines when and where the agent is required to perform the interaction task, and (3) an interaction policy that is a collection of subgoal policies that specialises in precise interaction tasks.

We first analyze each language instruction, and use the information to determine the basic interaction policies required to perform the task. Depending on the underlying composition, the control of the agent is alternated between (1) master policy and (2) various specialised policies for object interaction. Importantly, all interaction policies are compositional and independent, which allows formulating an instance specific policy execution sequence. In particular, we learn seven interaction policies, each of which specialises in a different sub-objective and can be integrated in a precise order to accomplish long-horizon planning tasks. We illustrate the overview of our proposed agent in Figure 2.

### 3.1 POLICY COMPOSITION CONTROLLER

For the task of interactive instruction following, the trajectories can be divided into several meaningful subgoals and detailed instructions are provided for each subgoal. Due to the requirement of long horizon planning for the task, we believe that it should be beneficial to first generate a high-level subgoal layout for the given task and then tackle each subgoal individually. To this end, we propose a policy composition controller (PCC), shown in Figure 2 which generates a sequence of subgoals $\mathcal{S} = \{s_i\}$ (where $s_i$ belongs to the set of subgaols defined by benchmark) for each 'step-by-step' instructions corresponding to the given task. The PCC's predictions correlate to semantic subgoals, subjecting the agent's logic to observation ('What is the agent attempting to accomplish right now?'), which enables us to monitor the progress of task completion by the agent. Specifically, we first encode the language instructions with a Bi-LSTM, followed by a self-attention module. The encoded step-by-step language instructions $\hat{x}_i$ are used as input for the PCC to generate the subgoal

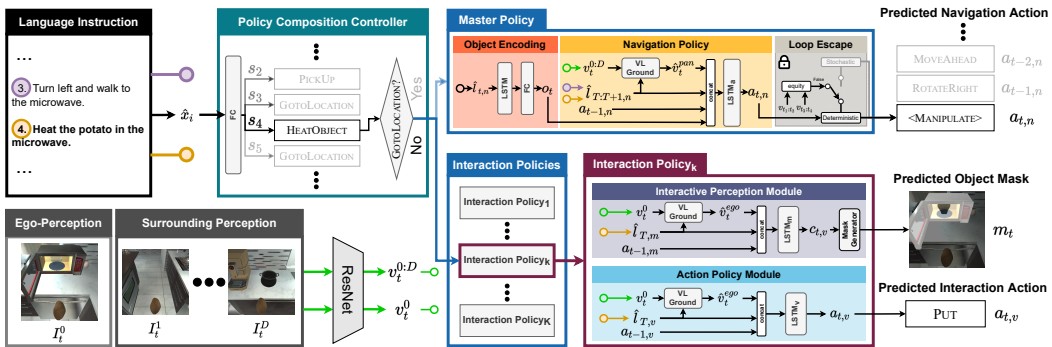

Figure 2: **Model Architecture.** The model architecture comprises of three main components (1) policy composition controller for defining the policy structure based on the given instructions, (2) a master policy (MP) responsible for navigation, and (3) a set of interaction policies (IP) specialising in the manipulation tasks. $I_t^d$ denotes an RGB frame from an explorable direction, $d \in [0, D]$, at the time step, $t$ where $d = 0$ indicates the egocentric direction. We encode $I_t^d$ using a pretrained ResNet and acquire a visual feature, $v_t^d$. $\hat{x}_i$ denotes each step-by-step instruction. $\hat{l}_{T,v}, \hat{l}_{T,m}$ denotes the encoded instruction for the 'interactive perception module' and 'action prediction module' respectively. $\hat{l}_{T:T+1,n}$ denotes the encoded 'subtask' instruction described in Sec. 3.2.1. $T$ refers to the index of the current subgoal. In our master policy, OEM outputs object encoding, $o_t$, using $\hat{l}_{T:T+1,n}$. 'VL-Ground' uses dynamic filters to capture the correspondence between visual and language features and outputs attended visual features, $\hat{v}_t^{pan}$ and $\hat{v}_t^{ego}$.

sequences. The agent must complete these subgoals in the specified order to accomplish the goal task. Formally, for each language instruction $\hat{x}_i$, the PCC defines a distribution over the subgoal action space as:

$$s_i = \arg\max_k(FC_1(\hat{x}_i)), \quad \text{where } k \in [1, N_{subgoals}], \tag{1}$$

where $FC_1$ denotes a single layer perceptron, and $N_{subgoals}$ denotes the number of subgoals *i.e.* 7. We train the PCC module in a supervised manner with the associated subgoal labels. On the ALFRED validation fold, the controller achieves 98.5% accuracy. Kindly refer Sec. A.3 for more details about these subgoals.

## 3.2 MASTER POLICY

We observed that the agent does not require information about the interactions it must perform for the given task when navigating in an environment. We require the agent to learn how to move from one interaction location to another with the help of language instruction. Inspired by these observations, we contend to use an independent module that can perform navigation while simultaneously marking the locations for object interaction along the way. For the task, we propose a master policy, which is illustrated by the upper-right block in Figure 2, for generating the navigational action sequence based on the multi-modal inputs. Let $\mathcal{A}_n$ denote the set of primitive navigation actions {MoveAhead, RotateRight, RotateLeft, LookUp, LookDown}. The master policy learns to navigate in the environment by learning a distribution over $\mathcal{A}_n \cup$ <MANIPULATE>, where <MANIPULATE> is the abstract token we introduced for the agent to signify when to move control to the next level of hierarchy, *i.e.*, the interaction policies, for completing manipulation sub-tasks. The master policy comprises of three basic modules: (1) an *object encoding module* that provides information about the object the agent needs to locate for interaction. (2) a navigation policy that outputs the navigation action sequence based on the multi-modal input for traversing the environment. (3) a loop escape module that helps the agent in escaping deadlock states.

### 3.2.1 SUBTASK LANGUAGE COMBINATION

The language instructions for the interactive instruction following task can be divided into two types: a) Navigation and b) Interaction. The agent needs to navigate to the necessary location and then perform interaction tasks, because of which the two types of instructions occur in pairs. To describe the complete task, the combination is repeated several times with varying locations and interaction

subgoals. For the better utilisation of the language input, we propose a new way of encoding step-by-step instructions for the navigation task.

Specifically, we define a subtask instruction as a combination of 'navigation to discover the relevant object' and the 'corresponding interactions'. For instance, in the subtask *Turn around and walk forward to the garbage bin on the floor by the TV. Pick up the blue credit card on the TV stand.*, the agent needs to interact with the credit card, which is a crucial information for the agent and also serves as a navigational criterion, *i.e.*, the agent should stop if it encounters the credit card in close vicinity. We observed that this information is missing in the first navigation command, but is present in the second interaction instruction which works as a motivation for the proposed combination. The navigation instruction provides the low-level information about the path to be traversed and the interaction instruction provides information about the object the agent needs to locate for appropriate interaction. We encode these subtasks using an instruction encoder comprised of a Bi-LSTM followed by a self attention module. $\hat{l}_{T:T+1,n}$ refers to the encoded feature for the combined subtask instruction corresponding to the navigation subgoal $T$.

### 3.2.2 Object Encoding Module

The task of instruction following can be broken down into two components, according to our observations: 1) locating the required object, and 2) interacting with the object. We propose a module that takes as input the subtask language instruction $l_{T:T+1,n}$ and outputs an encoding for the object the agent must locate for interaction. This guides the agent's navigation, and we recommend using it as a subgoal monitor to signify the completion of the current navigation subgoal.

*Navigation Subgoal Monitor.* The master policy learns to output an abstract token `<MANIPULATE>` along its navigation route, which passes control to a pretrained object detector (Mask RCNN) that validates whether the object provided by the OEM is present in the current frame or not. If the agent discovers the item, it switches to the appropriate interaction policy; otherwise, it continues with the navigation. The object encoder is composed of a Bi-LSTM, followed by a two layer perceptron which outputs the object class. For this object, the master policy utilises a learnable 100-dimensional embedding as input, as demonstrated in Eq. 2.

### 3.2.3 Navigation Policy

The navigation policy $\prod_n$ receives the processed multi-modal data as input and produces a sequence of actions as output. The architecture is based on the action prediction module of (Singh et al., 2021). It uses visual features, subtask instruction features (Sec. 3.2.1), and the embedding of the preceding time step action as inputs. Moreover, the object encoding supplied by the *object encoding module* (Sec. 3.2.2) is provided as input in order to further enhance the navigation capabilities of the agent.

The subtask instruction encoding $l_{T:T+1}$ provides low-level information relevant for navigation as well as the information about the object that the agent needs to interact with. This aids the agent in arriving at the correct location. To capture the relationship between the visual observation and language instructions, we dynamically generate filters based on the attended language features and convolve visual features with the filters, denoted by "VL-Ground" in Figure 2. To summarise, the LSTM hidden state $h_{t,n}$ of the master policy decoder, LSTM$_n$, is updated with four different features concatenated together as:

$$o_t = \underset{k'}{\operatorname{argmax}}(\text{FC}_o(\hat{l}_{T:T+1,n})) \quad where \; k' \in [1, N_{objects}]$$

$$h_{t,n} = \text{LSTM}_n([\hat{v}_t^{pan}; \; \hat{l}_{T:T+1,n}; \; a_{t-1,n}; \; o_t]) \tag{2}$$

$$a_{t,n} = \underset{k}{\operatorname{argmax}}(\text{FC}_n([\hat{v}_t^{pan}; \hat{l}_{T:T+1,n}; a_{t-1,n}; o_t; h_{t,n}])) \quad where \; k \in [1, |\mathcal{A}_n| + 1]$$

where $\hat{v}_t^{pan}$ denotes the attended visual features for the surrounding view as explained in Sec. A.2 in appendix at time step $t$, $\hat{l}_{T:T+1,n}$ denotes the attended subtask language features, $a_{t-1,n}$ represents the action given by $\prod_n$ in the previous time step, and $o_t$ denotes the object encoding given by the *object encoding module*. In addition, we use the subgoal progress and overall progress monitors similar to (Shridhar et al., 2020) for training the navigation policy. Details about the surrounding views is given in Sec. A.2 in the Appendix.

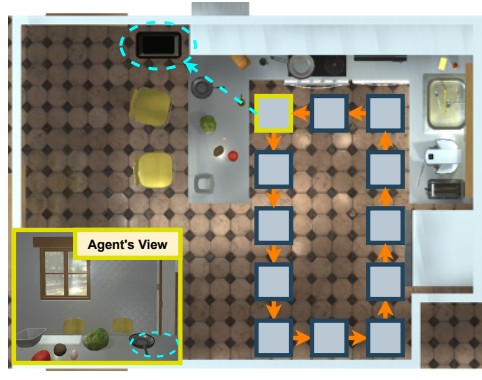 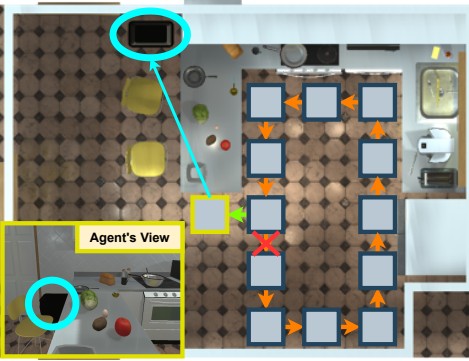

(a) Deadlock State    (b) Loop Escape Module

Figure 3: **Loop Escape Module.** The objective of the agent at the current time step is to move to a target object (a garbage can). (a) and (b) show an example of a deadlock state and the behavior of loop escape module when finding the target object, respectively. Each dark-blue square denotes the position of the agent. ○ denotes the target object that the agent should navigate to. → denotes the view direction of the agent. The dashed ○ and → indicate that the target object is invisible to the agent due to occlusion. → denotes actions taken by the agent a path taken in a deadlock state. In a deadlock state, the loop escape module cancels the current action that results in the deadlock state, ×, and takes a random navigation action, →.

### 3.2.4    LOOP ESCAPE MODULE

While navigating through an environment, the agent is prone to be caught around unanticipated obstructions such as tables or chairs, which might cause navigation failure. To circumvent such obstructions during navigation, (Singh et al., 2021) introduced obstruction evasion (OE). However, in the scenario where the agent predicts a repeated series of activities (i.e., a deadlock state), the agent may fail to navigate even without such barriers, resulting in limited exploration. The OE mechanism is unable to handle such challenges since repeated sequences of visual observations might not be recognised as obstacles based on visual similarity in consecutive frames. To address the problem, we propose a loop escape module, which allows for improved exploration at inference time by avoiding deadlock conditions. We use an external memory to memorise the history of visual observations, inspired by (Du et al., 2020). The agent identifies a deadlock state if any $t'$ exists that satisfies Equation 3, given a history of visual observations, $v_{0:t}$, up to the present time step, $t$.

$$v_{t'+w} = v_{t+w-W}, \quad \forall w \in [1, W], \tag{3}$$

where $W$ is a hyperparameter that indicates the length of the repeated actions. In the event of a deadlock, the stochastic policy executes a random navigation action to assist the agent in escaping.

Figure 3 depicts an example of a deadlock state and the behaviour of loop escape module in the case when the agent is unable to find a target object (a garbage can) due to occlusion (by the counter-top). As shown in Figure 3(a), the agent in a deadlock state tries to revisit previously explored areas, which may limit exploration. Loop escape module assists the agent in escaping the impasse condition, thus finding the garbage can, as depicted in Figure 3(b).

### 3.3    INTERACTION POLICIES

To abstract a visual observation to a consequent action, the agent requires a global scene-level comprehension of the visual observation whereas for localisation task, the agent needs to focus on both global as well local object specific information. Inspired by (Singh et al., 2021), we exploit separate streams for action prediction and object localization due to the contrasting nature of the two tasks, illustrated as 'Interaction Policy$_k$' in Figure 2.

The task require execution of varied subgoals with different levels of complexity. For instance, *Heat Subgoal* might require interaction with either a stove or a microwave whereas for *Pickup Subgoal* there is a variety of recepticles but the action sequence is simpler. To focus on individual sub-objectives, we train a module, $\prod_{m,k}$, for each subgoal where $k \in [1, N_{subgoals}]$. We observed that each interaction has its own properties and the navigation information history is irrelevant to

| Model | Input | | | Supervision | | Validation | | | | Test | | | |
|---|---|---|---|---|---|---|---|---|---|---|---|---|---|
| | *Vision* | *Language* | | *Vision* | *Language* | *Seen* | | *Unseen* | | *Seen* | | *Unseen* | |
| | RGB | Goal | Instr. | Depth | Subgoal | Task | GC | Task | GC | Task | GC | Task | GC |
| Seq2Seq | Single | ✓ | ✓ | ✗ | ✗ | 3.70 | 10.00 | 0.00 | 6.90 | 3.98 | 9.42 | 0.39 | 7.03 |
| LAV | Single | ✓ | ✗ | ✗ | ✗ | 12.70 | 23.40 | - | - | 13.35 | 23.21 | 6.38 | 17.27 |
| FPP | Single | ✓ | ✓ | ✗ | ✗ | 25.85 | 34.92 | 5.36 | 16.18 | 26.81 | 33.20 | 7.65 | 15.73 |
| EmBERT | Box | ✓ | ✓ | ✗ | ✗ | 37.44 | 44.62 | 5.73 | 15.91 | 31.77 | 39.27 | 7.52 | 16.33 |
| E.T. | Single | ✓ | ✓ | ✗ | ✗ | 33.78 | 42.48 | 3.17 | 13.12 | 28.77 | 36.47 | 5.04 | 15.01 |
| E.T. + synth. data | Single | ✓ | ✓ | ✗ | ✓ | **46.59** | 52.82 | 7.32 | 20.87 | 38.42 | 45.44 | 8.57 | 18.56 |
| LWIT | Box | ✓ | ✓ | ✗ | ✗ | 33.70 | 43.10 | 9.70 | 23.10 | 30.92 | 40.53 | 9.42 | 20.91 |
| HiTUT | Box | ✓ | ✗ | ✗ | ✓ | 25.24 | 34.85 | 12.44 | 23.71 | 21.27 | 29.97 | 13.87 | 20.31 |
| HLSM | Single | ✓ | ✗ | ✓ | ✓ | 25.00 | 36.37 | 11.80 | 24.70 | 25.11 | 35.79 | 16.29 | **27.24** |
| **HACR (Ours)** | Single | ✓ | ✓ | ✗ | ✓ | 45.96 | **54.55** | **19.96** | **38.19** | **40.66** | **47.62** | **16.70** | 26.17 |

Table 1: **Task and Goal-Condition Success Rate.** ✓ denotes the use of the corresponding attribute by the given approach and ✗ denotes the absence of the corresponding attribute. "Input" column represents the various input modalities used by the given approach. "Single" and "Box" under "RGB" column denotes that the visual features are either extracted from the entire image or bounding box predictions respectively. "Goal" and "Instr" columns indicates the use of high-level goal instruction and detailed step-by-step instructions by the given approach respectively. "Supervision" column provides details about the presence or absence of an additional supervision in training the corresponding approach apart from the primitive actions and object masks. "Depth" column denotes depth maps of RGB images and "Subgoal" denotes the high-level subgoal actions. We provide the highest values corresponding to each fold and each metric in blue and **bold**.

the task that makes it necessary to use an isolated hidden state for each interaction subgoal. More details about the architecture and training can be found in Sec. A.1 in the Appendix.

## 4 RESULTS AND DISCUSSION

First, we carry out a quantitative analysis of task success (Success Rate or SR) and goal condition success rates (Goal Condition or GC), and summarize the findings with prior approaches in Table 1. We compare our method with following prior-arts on ALFRED benchmark, Seq2Seq (Shridhar et al., 2020), LAV (Nottingham et al., 2021), FPP (Singh et al., 2021), EmBERT (Suglia et al., 2021), E.T. (Pashevich et al., 2021), LWIT (Nguyen et al., 2021), HiTUT (Zhang & Chai, 2021), and HLSM (Blukis et al., 2021).

HACR outperforms the preceding approaches on all metrics, as indicated in the table. Its ability to perform in unknown situations is evidenced by its increased success rate in unseen environments. For the validation fold, HACR outperforms the state-of-the-art (Blukis et al., 2021) with an improvement of 83.84 % in seen task SR, and 69.15 % in unseen task SR. For the test fold, HACR shows an improvement of 61.92 % in seen SR, and 2.5 % in in unseen SR. It indicates that our approach enhances the agent's overall grasp of all sub-tasks and their connection needed to complete a task successfully. We also discuss performance over individual subgoals in Table 6 in Appendix.

### 4.1 COMPARISON BETWEEN HIERARCHICAL AND FLAT POLICY AGENTS

We present the analysis of the learning process for the hierarchical and flat policy agents for seen and unseen environments. The performance of our hierarchical agent and the flat agent are compared quantitatively in relation to the number of iterations (expressed in terms of epochs), and the results are presented in Figure 4. As shown, hierarchical policy gives a major improvement over the flat policy. Its ability to perform in novel environments is evidenced by its increased success rate in unseen scenes. As depicted in Table 2 (#b), for seen and unseen task SR, the hierarchical agent outperforms the flat policy by 9.5% and 49.5%, respectively. In both seen and unseen 'Goal-Condition' SR, the hierarchical approach outperforms, with improvements of 8.3% and 29.1%, respectively. The greater performance of the hierarchical approach on both overall task success rate and goal condition suggests its comprehension of both short-term subtasks and long-horizon whole tasks.

Also, the hierarchical agent approach convergences significantly sooner than the monolithic agent (25th epoch vs 37th epoch), as shown in Figure 4a, demonstrating the computation-efficiency of our approach. Our policies go through two stages of training. We start by training the seven interaction

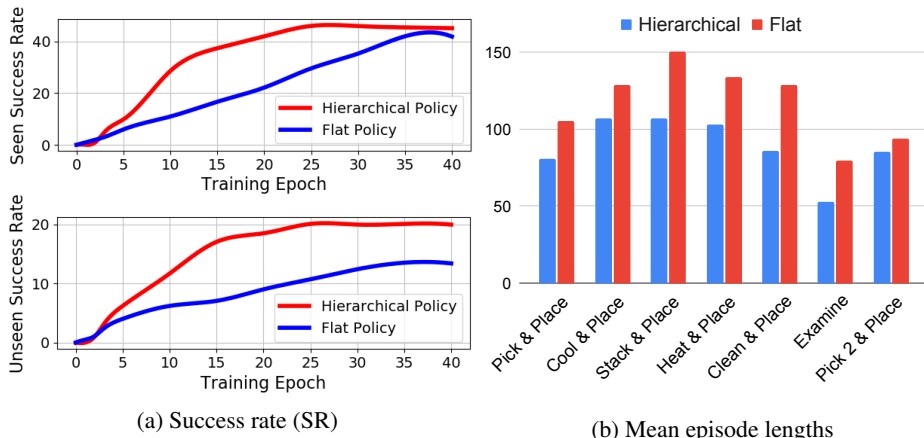

(a) Success rate (SR)                    (b) Mean episode lengths

Figure 4: **Hierarchical policy learns faster and more efficient action sequences.** Plot (a) shows the learning curves (success rates vs epochs) of the hierarchical and flat policy agents for unseen and seen environments. Plot (b) presents the average length of an episode traversed by a hierarchical or flat policy for the seven task types in ALFRED (Shridhar et al., 2020). The flat policy denotes the NIH ablated agent (#b) in Table 2.

policies, which collectively take two epochs to converge. We included them in the computation and began the hierarchical curve from the third epoch, which is the first epoch of our master policy.

Figure 4b represents the average length of a successful trajectory traversed by the hierarchical or flat policy agent, contrasting the efficiency of each agent. The hierarchical agent consist of the master policy that is dedicated solely to navigation, giving it a significant advantage over the flat policy agent that learns everything using the same network parameters. It was observed that due to the wide action space, the flat policy agent occasionally executes irrelevant interactions along the trajectory, which is not the case with HACR. The dedicated action sets for the master policy and interaction policies improve HACR by allowing the agent to avoid any unnecessary interactions while traversing to discover the desired object. The interaction policy modules also perform significantly better because they only master certain short-horizon tasks, which speeds up and simplifies the learning process. Table 6 in Appendix provides information on the subgoal performances for each module.

## 4.2 Interpretable Subgoal Modules

One of the benefits of the proposed approach is that it generates semantically meaningful subgoals. The generated subgoals are far more interpretable than a low-level action sequence. For instance, the low-level *put* action might be associated to any of the subgoals such as Heat, Cool, or Put. The high-level semantics are reasoned about by the hierarchical agent, and the agent's intent is considerably clearer. If the agent attempts to place the object in a refrigerator, it is most likely cooling it; if the receptacle is a microwave, it is attempting to heat it; and if the receptacle is a table, it is simply a pickup subgoal. In contrast, the flat policy agent considers it as a single atomic action regardless of the object and receptable. We present an example for more detail in Figure 7 in appendix.

## 4.3 Comparison between Modular and Non-Modular Policy Agents

In modular networks, the decision-making process is separated into numerous modules. Each module is designed to perform a certain function and is put together in a structure that is unique to each trajectory instance. Because of their compositional nature, such networks with the help of specialised modules often perform better in new environments than their flat counterparts (Hu et al., 2019; Blukis et al., 2019)

We present a quantitative comparison of interaction policies' modular structure (Table 2 #c). For this experiment, we train a single policy module to learn all interaction tasks. The decoupled pipeline for action and mask prediction, as well as the rest of the settings, are preserved. The modular agent outperforms the non-modular agent by 2.30% and 7.08% in seen and unseen task SR, respectively. It also performs significantly well in both seen and unseen 'Goal-Condition' criteria, with gains of

| | Components | | | Validation-Seen | | Validation-Unseen | |
|---|---|---|---|---|---|---|---|
| # | Modular interaction policy | Navigation interaction hier. | Object encoding module | Task | Goal-Cond. | Task | Goal-Cond. |
| - | ✓ | ✓ | ✓ | $45.96_{(1.1)}$ | $54.55_{(1.2)}$ | $19.96_{(0.2)}$ | $38.19_{(1.5)}$ |
| (a) | ✓ | ✓ | | $36.93_{(2.2)}$ | $42.22_{(4.1)}$ | $12.14_{(0.2)}$ | $28.43_{(3.1)}$ |
| (b) | ✓ | | ✓ | $41.96_{(2.1)}$ | $50.39_{(3.3)}$ | $13.35_{(0.2)}$ | $29.59_{(2.4)}$ |
| (c) | | ✓ | ✓ | $44.93_{(0.5)}$ | $53.38_{(1.1)}$ | $18.64_{(0.2)}$ | $34.98_{(1.5)}$ |
| (d) | | | ✓ | $28.96_{(1.9)}$ | $37.45_{(3.4)}$ | $7.32_{(0.3)}$ | $18.29_{(1.4)}$ |
| (e) | | ✓ | | $31.25_{(1.8)}$ | $39.64_{(2.9)}$ | $9.70_{(0.3)}$ | $22.34_{(1.3)}$ |

Table 2: **Ablation study for contributions of HACR.** For each metric, we report the task success rate. The absence of a checkmark (✓) denotes that the corresponding component is removed from HACR. Here, the 'Modular interaction policy' refers to the 7 subgoal modules, 'Navigation interaction hier.' denotes the hierarchy between navigation and interaction policy (Sec. 3), and 'Object encoding module' is introduced in 3.2.2. Details are presented in Sec. 4.1, 4.3, and 4.4. We report average over 5 runs (with random seeds) with standard deviations depicted in sub-script parenthesis

2.19% and 9.17%, respectively. The greater performance of the modular policy in both task and goal-condition metrics highlights the benefits of modular structure in long-horizon planning tasks.

## 4.4 ABLATION STUDY

We conduct a series of ablation analysis on the components of HACR and report the results in Table 2 to evaluate the significance of each module with empirical studies.

**(a) Object encoding module (OEM).** First, we investigate the ablation of Object Encoding Module. We remove the navigation subgoal monitor and train the navigation policy without object information. The agent can complete some objectives, but it lacks object information, which functions as a stopping criterion, preventing it from navigating properly. Therefore, it is unable to completely comprehend the relationship between the step-by-step instruction and the visual trajectory, limiting the agent's capacity to explore and connect various interaction policies required for the task's completion, thus resulting in a huge performance drop.

**(b) Navigation interaction hierarchy (NIH).** Next, we demonstrate the importance of hierarchy between navigation and interaction policies. For this ablation, we utilize the same network for learning the navigation and interaction action prediction. For target class prediction, we preserve the IPM. To eliminate the benefit of the language distribution, we use the concatenation of all step-by-step instructions as the language input and conduct action and mask prediction while leaving the other modules unaltered. The ablated model's task success rates drop dramatically, showing that it is unable to effectively utilise the available inputs.

**(d) and (e) Individual Components** Next, we provide the individual performance of the two major components of our framework which brings the most empirical gain, OEM and NIH in row (d) and (e) respectively in the absence of other components. Even in the lack of hierarchy and modular structure, the agent (d) has significantly higher performance than (Singh et al., 2021), highlighting the relevance of target object information for navigation and the effectiveness of our OEM. The lower performance of agent (e) denotes that the hierarchical architecture requires the inclusion of other components for optimal performance. The agent's performance increases when these components are combined (#c vs (#d and #e)) indicating that the components are complimentary.

## 5 CONCLUSION

We address the problem of interactive instruction following. To effectively tackle the long horizon task, we propose Hierarchical Approach for Compositional Reasoning (HACR), that decomposes the task into high-level subgoals and accomplishes each subgoal with a specialized submodule. We propose a policy composition controller (PCC) to disentangle the task into high-level subgoals. For improving the navigation performance, we propose an object encoding module (OEM) to explicitly encode target object information during internal state update. We also propose a loop escape module to circumvent the deadlock states during navigation, for better exploration. Our agent outperforms prior arts by significant margins and achieves a new state-of-the-art.

ETHICS STATEMENT

The embodied AI research will expedite the deployment of AI systems to help humans by its versatility of adapting to a new environment out of the factory or research labs. As all embodied AI, however, would suffer from adversarially annotated data as well as data bias in a form of language instructions, which may cause ethnic, gender or biased gender issues, the proposed method would not be an exception. Although the proposed method has *no intention* to allow such problematic cases, the method may be exposed to such threats. Relentless efforts should be made to develop mechanisms to prevent such usage cases in order to make the robotics AI safer and enjoyable to be used by humans.

REPRODUCIBILITY STATEMENT

We take the reproducibility of research very seriously and describe the algorithm in detail in Sec. 3 and Sec. A.1. We also solemnly promise to release all codes, environment information, learned models, and complete task configurations in a public repository.

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

# A APPENDIX

## A.1 DETAILS FOR INTERACTION POLICY

Let $\mathcal{A}_m$ denote the set of primitive interaction actions {Open, Close, ToggleOn, ToggleOff, Pickup, Put}. The APM architecture consist of an LSTM decoder that takes as input the attended language encoding $\hat{l}_{T,m}$ corresponding to the current manipulation subgoal, the egocentric visual features $\hat{v}_{t,m}^{ego}$ and the previous time step action $a_{t,m}$ and outputs a distribution over the manipulation action space $\mathcal{A}_m \cup \texttt{<STOP>}$. The manipulation policy uses $\texttt{<STOP>}$ token as an indicator for completion of the interaction task and shifts the control to the master policy for further task completion.

Formally:

$$
\begin{aligned}
h_{t,m}^{apm} &= \text{LSTM}_m([\hat{v}_{t,m}^{ego}; \hat{l}_{T,m}; a_{t-1,m}]), \\
a_{t,m} &= \underset{k}{\arg\max}(\text{FC}_m([\hat{v}_t^{ego}; \hat{l}_{T,m}; a_{t-1,m}; h_{t,m}^{apm}])) \quad where \ k \in [1, |\mathcal{A}_m| + 1].
\end{aligned}
\tag{4}
$$

The IPM uses the same inputs as APM and outputs the object class that the agent needs to interact with at the current time step. Similar to (Singh et al., 2021), we use a pretrained Mask-RCNN to obtain the interaction mask corresponding to the object class. For more details about the architecture, kindly refer to (Singh et al., 2021).

$$
\begin{aligned}
h_{t,v}^{vpm} &= \text{LSTM}_v([\hat{v}_{t,v}^{ego}; \hat{l}_{T,v}; a_{t-1,v}]), \\
c_{t,v} &= \underset{k}{\arg\max}(\text{FC}_v([\hat{v}_t^{ego}; \hat{l}_{T,v}; a_{t-1,v}; h_{t,v}^{vpm}])) \quad where \ k \in [1, N_{objects}].
\end{aligned}
\tag{5}
$$

We train our interaction policies in a supervised manner using imitation learning from expert groundtruth trajectories. Specifically, each interaction policy is trained to minimize the cross entropy loss as:

$$
\mathcal{L}_{int} = \sum_t a_{t,m}^* \log p_m + \sum_t c_{t,v}^* \log p_v,
\tag{6}
$$

where $p_m = \text{FC}_m([\hat{v}_t^{ego}; \hat{l}_{T,m}; a_{t-1,m}; h_{t,m}^{apm}])$ and $p_v = \text{FC}_v([\hat{v}_t^{ego}; \hat{l}_{T,v}; a_{t-1,v}; h_{t,v}^{vpm}])$ denotes the probability distribution of action and class prediction, respectively, and $a_{t,m}^*$ and $c_{t,v}^*$ denotes the corresponding ground truth actions and classes obtained from given expert trajectories, respectively.

## A.2 SURROUNDING PERCEPTION

HACR investigates and perceives surrounding views to improve the agent's perception and gain a better comprehension of the environment. Let $v_t^d$ be a visual feature from each navigable direction, $d \in [0, D]$, at the time step, $t$, where $D$ represents the number of navigable directions and $d = 0$ the agent's egocentric direction. After gathering additional observations, $v_t^{0:D}$, we acquire attended visual features, $\hat{v}_t^{pan}$, using the dynamic filters and concatenate all attended visual features as:

$$
\hat{v}_t^{pan} = [\hat{v}_t^0; \hat{v}_t^1; \cdots; \hat{v}_t^D].
\tag{7}
$$

Using the attended visual features, we update the master policy decoder and predict masks and actions as described in Equation 2.

## A.3 EXPERIMENTAL DETAILS

**Dataset and Metrics.** For training and evaluation, we used the ALFRED benchmark (Shridhar et al., 2020). This benchmark provides expert trajectories for the agents performing household tasks (e.g. Boil an egg) in a simulated environment on AI2-THOR (Kolve et al., 2017). The dataset is divided into three parts; 'train', 'validation', and 'test' set. To evaluate the generalisation ability of an embodied agent to novel environments, the benchmark further divides 'validation' and 'test' trajectories into *seen* and *unseen* splits. Unseen comprises a set of rooms which are held out during training and scenes that are exposed to the agent during the training are termed as *seen*.

The benchmark provides a high-level Goal Statement describing the final task and several low-level Step-by-Step Instruction describing each subgoal that needs to be accomplished in the given order for successful completion of goal task. The benchmark has 7 types of subgoals; GOTOLOCATION, PICKUPOBJECT, PUTOBJECT, COOLOBJECT, HEATOBJECT, CLEANOBJECT, SLICEOBJECT, and TOGGLEOBJECT. GOTOLOCATION denotes navigation subgoal where the agent requires to locate a target object for interaction. PICKUPOBJECT requires the agent to pick up an object. PUTOBJECT requires the agent to put an object onto a receptacle. COOLOBJECT, HEATOBJECT, and CLEANOBJECT requires the agent to cool, heat, and clean an object while interacting with the required receptacle (*e.g*., Fridge, Microwave, SinkBasin, etc.). SLICEOBJECT requires the agent to slice an object using a knife, resulting in sliced objects. TOGGLEOBJECT requires the agent to toggle on or off an object such as a floor lamp.

Following the common evaluation protocols (Blukis et al., 2021), we evaluate our agent on two metrics. Success rate (SR) provides the percentage of tasks that had all successful goal conditions. Goal condition rate (GC) is the percentage of satisfied goal-conditions across all tasks.

**Training and Evaluation.** We begin by assessing the performance of each subgoal policy in relation to its specific task. On the ALFRED validation set, we present the success rate for each subgoal in Table 6. To accomplish this, we employ the expert trajectory to guide the agent through the episode until it reaches the subgoal task. The agent then uses the task specific policy to infer based on the language directive and current visual frame. Following that, we train and evaluate the master policy's performance during independent behaviour cloning on expert trajectories, i.e. assuming oracle subgoal-policies. With ground truth subgoal sequences, the master policy is able to generalise effectively and achieves 29.96% success rate (SR) on unseen validation data. Then we train the Object Encoding Module and the Policy Composition Controller in isolation and combine all the modules together as described in Sec. 3.

**Implementation Details.** Visual features are encoded with a pretrained ResNet-18 (He et al., 2016) after resizing input images to $224 \times 224$. To perceive surrounding views, we additionally gather visual observations from four navigable directions (i.e., ROTATELEFT, ROTATERIGHT, LOOKUP, and LOOKDOWN). We train each module of our approach independently using the Adam Optimizer with the initial learning rate of $10^{-3}$. We augment visual features by shuffling the channel order of each image (Singh et al., 2021) and applying predefined image augmentations (Cubuk et al., 2019). We set the coefficients for the subgoal and overall progress monitor objectives as 0.2, and 0.3 respectively. The value of $W$ for the loop escape module is set as 10 using a grid search on validation set.

## A.4 FURTHER ABLATION STUDY

### A.4.1 NETWORK COMPONENTS ABLATION

| | Components | | | Validation-Seen | | Validation-Unseen | |
|---|---|---|---|---|---|---|---|
| # | Object-centric mask prediction | Factorised interaction policy | Data Augmentation | Task | Goal-Cond. | Task | Goal-Cond. |
| - | ✓ | ✓ | ✓ | $45.96_{(1.1)}$ | $54.55_{(1.2)}$ | $19.96_{(0.2)}$ | $38.19_{(1.5)}$ |
| (a) | ✓ | ✓ | | $40.18_{(1.1)}$ | $45.51_{(2.0)}$ | $14.91_{(0.5)}$ | $34.79_{(2.3)}$ |
| (b) | ✓ | | ✓ | $27.15_{(2.4)}$ | $38.50_{(2.2)}$ | $11.11_{(0.2)}$ | $33.81_{(2.9)}$ |
| (c) | | ✓ | ✓ | $9.27_{(0.5)}$ | $20.95_{(1.1)}$ | $2.56_{(0.1)}$ | $22.67_{(1.3)}$ |

Table 3: **Ablation study for design components of HACR.** For each metric, we report the success rate. Checkmark denotes that the corresponding component is used. Please refer to Sec. A.4.1 for details about each component. We report average over 5 random runs with standard deviations in sub-script parenthesis

**(a) Data augmentation (DA).** We start by ablating the data augmentations (described in Sec. A.3 which lowers the performance across the board, highlighting its usefulness in training a smarter agent by reducing imitation learning sample complexity. This results in a performance drop of 12.6% and 25.0% in the seen and unseen task success rate respectively.

**(b) Factorised interaction policies (FIP).**   Then, under the interaction policies, we replaced the decoupled pipeline with a unified pipeline (Singh et al., 2021) for both action sequence prediction and interaction mask generation. Due to shared processing for mask and action modules, the ablated model's performance drops substantially, showing its inability to effectively utilize the language information.

**(c) Object-centric mask prediction (OCMP).**   Finally, we train the interaction policies without the use of object based mask prediction which uses a Mask-RCNN as described in Sec. A.3. Similar to (Shridhar et al., 2020), we directly upsample the joint embedding using deconvolution layers to generate the interaction mask. We see a significant decline in performance on both seen and unseen folds due to poor mask generating ability, emphasising the need of object information for consistently accurate mask formation.

| Input Ablations | Validation-Seen | | Validation-Unseen | |
|---|---|---|---|---|
| | Task | GC | Task | GC |
| All inputs (HACR) | 45.96 | 54.55 | 19.96 | 38.19 |
| Egocentric View Only | 30.48 | 36.13 | 12.42 | 23.72 |
| No Vision | 0.61 | 6.87 | 0.24 | 8.25 |
| No Language | 3.65 | 10.66 | 1.21 | 7.78 |
| Goal-Only | 8.53 | 16.50 | 2.07 | 10.04 |
| Single Navigation Instruction | 12.19 | 23.71 | 3.04 | 14.95 |
| w/o Loop Escape Module | 45.43 | 54.18 | 19.88 | 37.98 |

Table 4: **Input Ablations and LEM Ablation.** We ablate different inputs to our hierarchical network and report the task success rates and GC success rates. Kindly refer Sec. A.4.2 for further details

### A.4.2   INPUT ABLATION

To further investigate our agent's vision and language bias, we ablate the inputs to our model in Table 4.

**Egocentric View only.**   This ablation specifies the condition under which the agent is deprived of its surrounding views (Sec A.2) while navigating without affecting any other module. The agent continues to perform admirably and outperforms the HLSM (Blukis et al. (2021)) on the validation fold by a large margin in seen trajectories, while also providing a significant improvement in task success rate in unseen environment. This demonstrates the significance of peripheral perceptions in navigating a novel environment.

**No Language.**   We discovered that when the agent receives only visual inputs without the language instructions, it is still capable of doing some tasks by memorising common sequences and target classes in the 'seen' fold. but does not adapt to the unseen environment.

**No Vision.**   This ablation demonstrates that the agent can still perform some subgoals, mostly navigation, by following the instructions to move along a few short trajectories, but not able to complete goal tasks because it involves mask generation (object selection), which is entirely dependent on visual observation.

**Goal-Only.**   We define the setting where both the navigation and interaction policies receive only the goal task information. It gives a general idea of the task but doesn't contain any information about the tedious action combinations the agent needs to perform to get the task done. This leads to a significant depletion in the task success rate.

**Single Navigation Instruction.**   This is when the master policy only receives the navigation instruction corresponding to the current navigational subgoal. The agent is able to somewhat navigate but doesn't have the object information that acts as the stopping criterion which hinders the agent's ability to perform interaction at the correct place in the environment. This demonstrates the benefit of our object-centric navigational approach.

A.5    ADDITIONAL RELATED WORKS

In this section we extend our realated works, to provide a detailed technical comparison with the prior arts involving hierarchical frameworks (Blukis et al., 2021; Zhang & Chai, 2021).

**Comparison with HLSM (Blukis et al., 2021).** Here we compare the technical aspects of HACR and HLSM. Both the works uses action hierarchy to decompose trajectory into high-level subgoal actions and subgoals into low-level action sequences. There are differences between the subgoals anticipated by HACR and HLSM. HLSM only employs interaction subgoals and considers navigation to be part of the same subgoal in order to reach the place of interaction, whereas HACR includes an extra navigation subgoal 'GoTo(object)' in which the agent navigates the environment to discover the object of interest. HACR also contains the compound subgoals for Clean and Heat , whereas HLSM would forecast the complete sequence of primary subgoals like Put, Pick, Toggle etc.

In terms of model architecture, HLSM defines the task layout by using a hierarchical model with high-level and low-level controllers. The navigation and interaction components of a particular subgoal are collaboratively worked on by the low-level controller. HACR, on the other hand, adopt a three-level hierarchy with independent modules for each level, namely, PCC, which provides the high-level structure of the trajectory, MP, which is responsible for navigation, and IP, which is composed of modules specialised for each interaction job.

We believe that HLSM and HACR are complementary because HLSM uses depth supervision to estimate the 3D layout of the environment for detailed visual understanding while our HACR exploits high-level language instructions for action planning. The depth supervision could be complementary to low-level instructions because the supervision of HLSM provides rich visual information by depth, which contains visual details, while the supervision of our HACR provides abstract language information for high level guidance for actions. We argue that it is non trivial to claim which supervision is less or more. Using the Object Encoding Module, HACR also adds a navigational subgoal monitor that checks the availability of object in the scene before executing the interaction policy.

A possible reason that HACR shows strong performance in seen but less in unseen is insufficiency of data. Due to the limited data size for imitation learning of the given benchmark (note that despite its size being largest in the literature, it is an offline dataset, *i.e.*, the training data are obtained from the simulator by *a single expert trajectory* not using the simulator to learn various possible scenarios) and large language domain gap in seen/unseen environments, generalization is challenging. As the detailed supervision about the environment, e.g., depth information, may help generalization (HLSM), using both supervision for environments and detailed language would reduce the generalization gap further.

**Comparison with HiTuT (Zhang & Chai, 2021).** We also compare the technical aspects of HACR and HiTuT. HiTut also employs action hierarchy to decompose trajectory into high-level subgoal actions and subgoals into low-level action sequences like HACR and HLSM. Even though HiTuT divides the task into a hierarchy of high-level subgoals and low-level actions, they utilize a unified transformer model with multiple heads to predict the subgoals and actions. In contrast, HACR consists of independent modules for subgoal planning, navigation and varying interaction tasks. The modular design allows for separate processing via specialised modules, making the learning process simpler due to short-term reasoning and enables easy inclusion of additional interaction subgoals.

A.6    ADDITIONAL EXPERIMENTAL RESULTS

There are seven high-level task categories in the ALFRED benchmark (Shridhar et al., 2020). Table 5 presents the quantitative comparison of our approach with a flat policy, a seq2seq model (Shridhar et al., 2020), and prior hierarchical baselines (Blukis et al., 2021; Zhang & Chai, 2021) for each task type. Blukis et al. (2021) outperforms our agent on short-horizon tasks like 'Pick & Place' and 'Examine.' The two most complex and longest-horizon task categories in ALFRED are 'Stack & Place' and 'Pick Two & Place.' On these task types HACR gives an unseen success rate of 13.8% and 22.2%, respectively, compared to HLSM's 1.8% and 12.3%, and outperforms all baselines in 5 of the 7 task types in unseen SR. Overall, HACR outperforms all the baselines in terms of average seen and unseen success rates. This is inline with the detailed analysis in Sec. 4.3 and 4.1 which describes the effectiveness of our approach to long-horizon trajectories.

| Task-type | Seq2Seq | | Flat policy | | HiTuT | | HLSM | | HACR (Ours) | |
|---|---|---|---|---|---|---|---|---|---|---|
| | Seen | Unseen | Seen | Unseen | Seen | Unseen | Seen | Unseen | Seen | Unseen |
| Pick & Place | 7.0 | 0.0 | 57.2 | 5.2 | 35.9 | 26.0 | 42.3 | **28.0** | **62.7** | 8.0 |
| Cool & Place | 4.0 | 0.0 | 40.9 | 19.9 | 19.0 | 4.6 | 11.9 | 3.7 | **46.0** | **31.2** |
| Stack & Place | 0.9 | 0.0 | 23.2 | 10.3 | 12.2 | 7.3 | 8.7 | 1.8 | **24.3** | **13.8** |
| Heat & Place | 1.9 | 0.0 | 47.7 | 11.4 | 14.0 | 11.9 | 12.1 | 0.0 | **52.3** | **16.9** |
| Clean & Place | 1.8 | 0.0 | 38.8 | 24.6 | **50.0** | 21.2 | 25.0 | 10.6 | 43.7 | **38.1** |
| Examine | 9.6 | 0.0 | 48.5 | 10.3 | 26.6 | 8.1 | 37.2 | **22.5** | **52.1** | 13.9 |
| Pick Two & Place | 0.8 | 0.0 | 35.0 | 14.9 | 17.7 | 12.4 | 35.5 | 12.3 | **39.5** | **22.2** |
| Average | 3.7 | 0.0 | 41.6 | 13.8 | 25.1 | 13.1 | 24.7 | 11.3 | **45.9** | **19.9** |

Table 5: **Success rates across seven task types in ALFRED.** 'Seq2Seq' refers to the method by Shridhar et al. (2020). 'Flat policy' refers to Sec. 4.4 (b). Here HiTuT (Zhang & Chai, 2021) and HLSM (Blukis et al., 2021) denotes the prior arts involving hierarchical aproach. We indicate the highest values corresponding to each task type in **blue** and all numbers are in percentage. We evaluate the agents on the **Validation** set.

Table 6 presents the performance of individual subgoal policies on corresponding subgoals. Our hierarchical agent relatively outperforms the flat policy by 50% and 21% average on unseen and seen validation set respectively. The modular nature of our interaction policies allows them to learn specific short horizon subgoal tasks as illustrated in Figure 8-14, which reduces the complexity and makes the learning faster and easier.

| Task-type | Seq2Seq | | Flat policy | | HACR (Ours) | |
|---|---|---|---|---|---|---|
| | Seen | Unseen | Seen | Unseen | Seen | Unseen |
| Pickup | 32 | 21 | 66 | 39 | **82** | **68** |
| Put | 81 | 46 | 73 | 45 | **90** | **68** |
| Cool | 88 | 92 | 77 | 90 | **95** | **94** |
| Heat | 85 | **89** | 69 | 49 | 85 | 87 |
| Clean | 81 | 57 | 65 | 51 | **81** | **73** |
| Slice | 25 | 12 | 65 | 49 | **73** | **79** |
| Toggle | 100 | 50 | 79 | 30 | **97** | **53** |
| Average | 70 | 46 | 71 | 50 | **86** | **75** |

Table 6: **Subgoal success rate.** The highest values per fold and task are shown in **blue**.

### A.6.1 ABLATION STUDY ON VISUAL INPUT FOR POLICY COMPOSITION CONTROLLER

The impact of visual features on PCC is presented in the Table 7. The agent does not gain from accessing visual information, as evidenced by the drop in performance for PCC w/ vision. We believe that this happens due to the 'causal misidentification' phenomenon (de Haan et al., 2019); when receiving more information (*e.g.*, RGB observation), the agent could learn to exploit irrelevant information (*e.g.*, a chair in the room) to a target task (*e.g.*, Heat a potato) rather than necessary information (*e.g.*, Microwave and potato), leading to performance drop.

| Input Ablations | PCC | **Validation-Seen** | | **Validation-Unseen** | |
|---|---|---|---|---|---|
| | | Task | GC | Task | GC |
| PCC w/o vision | **98.5** | **45.96** | **54.55** | **19.96** | **38.19** |
| PCC w/ vision | 96.8 | 45.12 | 53.01 | 19.49 | 37.59 |

Table 7: **Input Ablations for policy composition controller (PCC).** "PCC w/o vision" denotes the original HACR agent, where as "PCC w/ vision" denotes the agent using additional visual input for Policy Composition Controller (PCC) The highest values per fold and task are shown in **blue**. The second column depicts the performance of PCC for predicting the subgoal sequence evaluated on validation set.

### A.6.2 ABLATION STUDY ON VISUAL INPUT FOR MASTER POLICY

The impact of visual features on MP is presented in the Table 8. The MPP w/o vision demonstrates that the agent can still complete a few goal conditions, mostly the first navigation instruction, by

following the instructions to move along a few short trajectories, but not able to complete goal tasks because it involves interaction with objects in the environment i.e. localising objects, which is entirely dependent on visual observation.

| Input Ablations | Validation-Seen | | Validation-Unseen | |
|---|---|---|---|---|
| | Task | GC | Task | GC |
| MP w/ vision | 45.96 | 54.55 | 19.96 | 38.19 |
| MP w/o vision | **1.58** | **12.57** | **0.73** | **10.27** |

Table 8: **Input Ablations for Master Policy (MP).** "MP w/ vision" denotes the original HACR agent, where as "MP w/o vision" denotes the agent without visual input for Master Policy (MP). The highest values per fold and task are shown in **blue**.

### A.6.3 SENSITIVITY ANALYSIS FOR LOOP ESCAPE MODULE (LEM) WINDOW ($W$)

The impact of the value of $W$ for Loop Escape Module (LEM) is examined in Figure 5 by conducting a grid search. We observe that our agent achieves the best performance with $W = 10$, which indicates the efficacy of LEM. Note that the performance does not drastically decrease by other values of $W$, which indicates that it prevents the agent from performance degradation by other values of $W$.

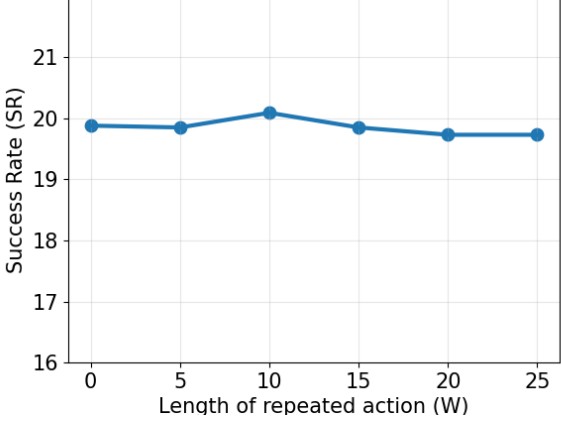

Figure 5: **The sensitivity curve for the value of $W$ for loop escape module (LEM).** Y-axis denotes the success rate for the unseen validation split.

### A.7 WHAT TO LEARN? VS. WHAT TO BAKE IN AS AN INDUCTIVE BIAS?

We agree that our method is a more or less 'classical' pipelined approach and has many carefully designed components as a set of inductive biases. We attribute our design to (1) lack of sufficient training data and (2) requirement of heavy computational cost. (1) Given our framework is the behavior cloning (BC) (Bain & Sammut, 1995), though the benchmark we used is the largest in the literature, the rigid nature of the BC approach (*i.e.*, any deviation from the expert trajectory over time is not allowed to learn a behavior) requires vast amount of training examples (*i.e.*, may be not sufficient to be the largest dataset in the literature) to learn a *satisfactory* model. (2) To overcome the rigidity of learning a BC model, there are many approaches proposed in the literature (Ho & Ermon, 2016; Fu et al., 2017). But they are mostly computationally expensive (Kostrikov et al., 2018). We aim the niche between both problems by designing well crafted components for better encoding the given dataset even in the rigid BC framework.

Similar to our answer to the previous question, our inductive bias could be useful for this task not just for this benchmark by the analogy of convolutional neural networks for images. By the universal approximation theorem, the 2D information, *e.g.,* images, could be encoded by a multi-layer

perceptron (MLP) with sufficient amount of data. Given that it is difficult to fathom the sufficient amount of data to train the MLP, we have a very successful design choice for encoding 2D information; the patch wise encoding and convolutional operations for its processing. Now, with 1M images, *e.g.*, the ImageNet-1K, we can have a decently performing image classifier. We hope that our designed components could be useful to inspire the future research in designing a data driven model to address the Embodied AI task successfully. In sum, we pose our work as a stepping stone for designing and learning a successful data driven model in the future.

### A.8 Does separate networks increase hardware/training/computational cost?

In this section, we provide a discussion about the hardware, training, and computational cost when learning and exploiting separate networks for each subgoal.

#### A.8.1 Hardware cost

Our model would increase the memory cost linearly by the number of subgoals (0.23GB per subgoal) because models for each policy should be loaded in the memory proportional to the number of subgoals. Considering the large sized memory in modern computer systems (*e.g.*, usually larger than 8GB for PC and 6GB for high-end mobile phones), the additional memory cost may not be considered large, but it is certainly a cost.

#### A.8.2 Training cost

The training cost for each policy may marginally or may not increase because we divide a task into multiple subgoals and learning the specialized model for each subgoal takes a few epochs as the training set for each policy is *easier* (*i.e.*, sharing similar semantics) for training than the data for the *flat model*. In Figure 4 (a) in the revision, we show that learning the specialized networks requires less training epochs compared to learning a unified network. Therefore, even though there are multiple subgoals to be learned, the total training cost would increase marginally (and even embarrassingly parallelizable) or even decrease thanks to the reduced training cost per each subgoal added to an overhead of predicting subgoals.

#### A.8.3 Computational cost

The computational cost may marginally increase or even decrease because specialized networks for subgoals could infer each subgoal efficiently by reducing unnecessary exploration. Figure 4 (b) in the revision shows that the specialized networks require shorter mean episode lengths to accomplish tasks. With the efficient subgoal achievement, the overall computational cost at inference would be similar or even decrease, compared to the model with the flat policy learning.

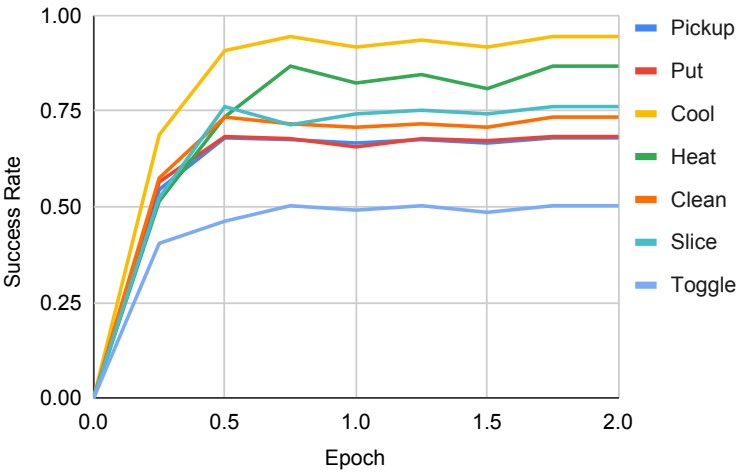

Figure 6: Plot shows the learning curve (subgoal success rates vs epochs) for the seven subgoal policies described in Sec. A.1

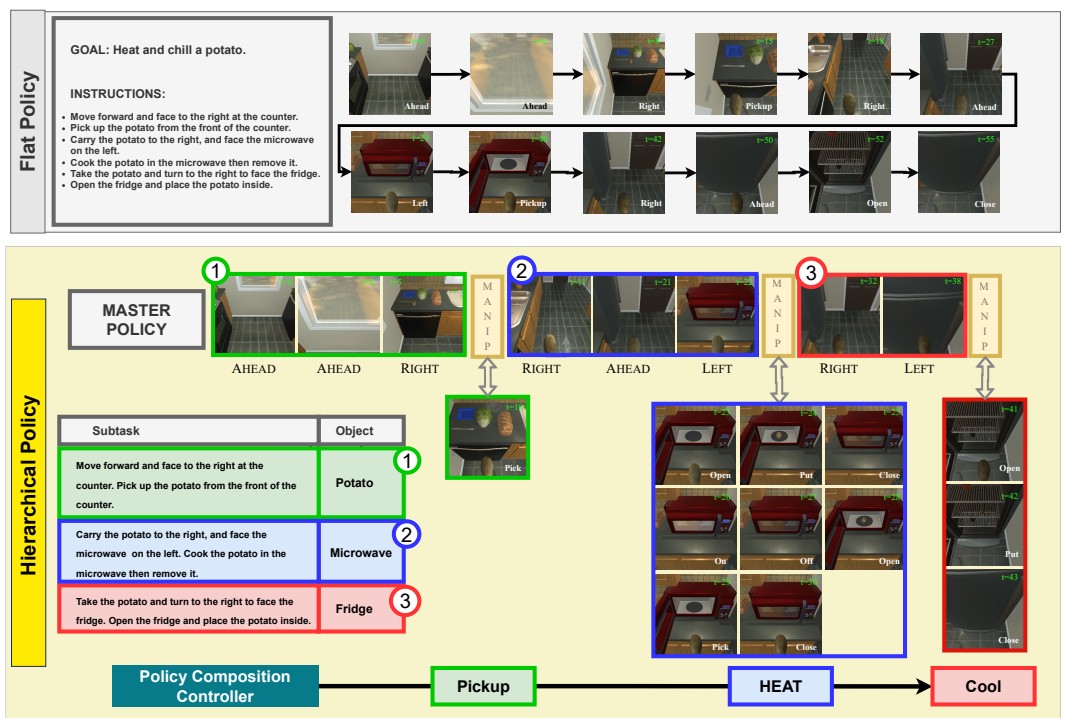

Figure 7: **Inference of the task 'Heat and Chill a potato.' using hierarchical policy vs. using flat policy.** The hierarchical agent decompose the instructions into subtasks each including a navigation and an interaction subgoal. The PCC at the bottom outputs a sequence of subgoals used for selecting the appropriate interaction policy. The object column demonstrates the output of the OEM. The master policy uses the subtask instruction, object encoding, and visual observation to infer the navigation action at every time step as shown at the top. It predicts the <MANIPULATE> token to shift the agent's control to the interaction policies along the way. On completion of the interaction task, the control is shifted back to the navigation policy. The hidden state of the navigation policy is keeps a continuous encoding of navigation history without any interference of the interaction policy whereas each interaction policy uses an independent hidden state. The flat policy shown at the top, on the other hand shares the same parameters for learning both navigation and interaction subtasks. This disentanglement results in simpler and faster learning by enabling parallel computation for various policies.

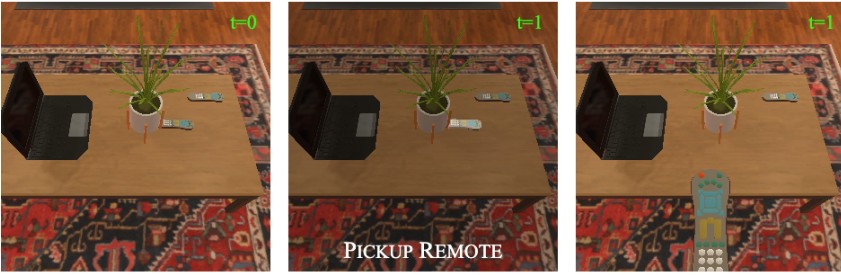

Figure 8: Pickup Object interaction policy agent completes a subgoal task 'Take the remote.' in an unseen environment.

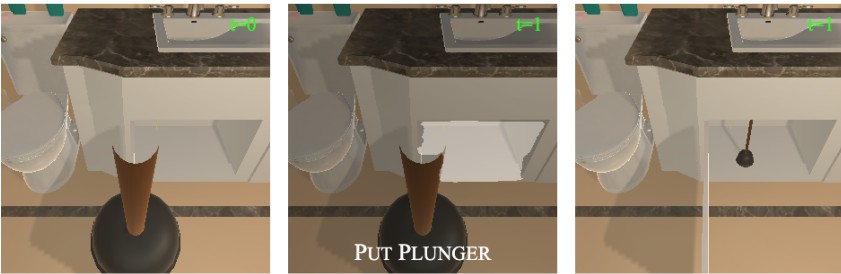

Figure 9: Put Object interaction policy agent completes a subgoal task "Put the plunger below the sink." in an unseen environment.

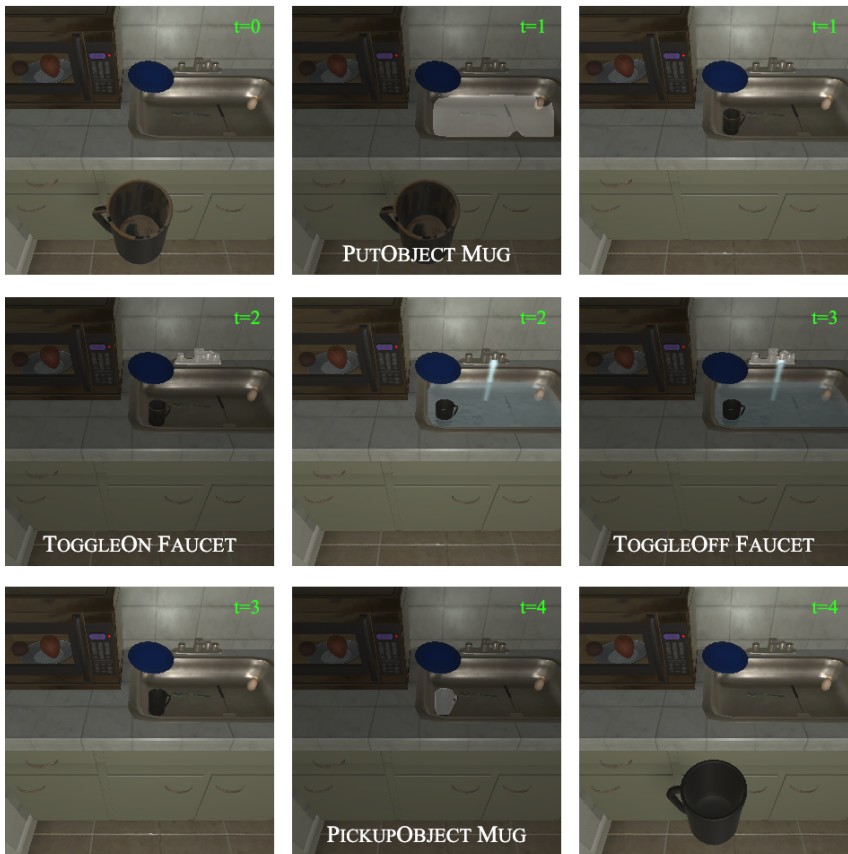

Figure 10: Clean Object interaction policy agent completes a subgoal task "Clean the mug in the sink." in an unseen environment.

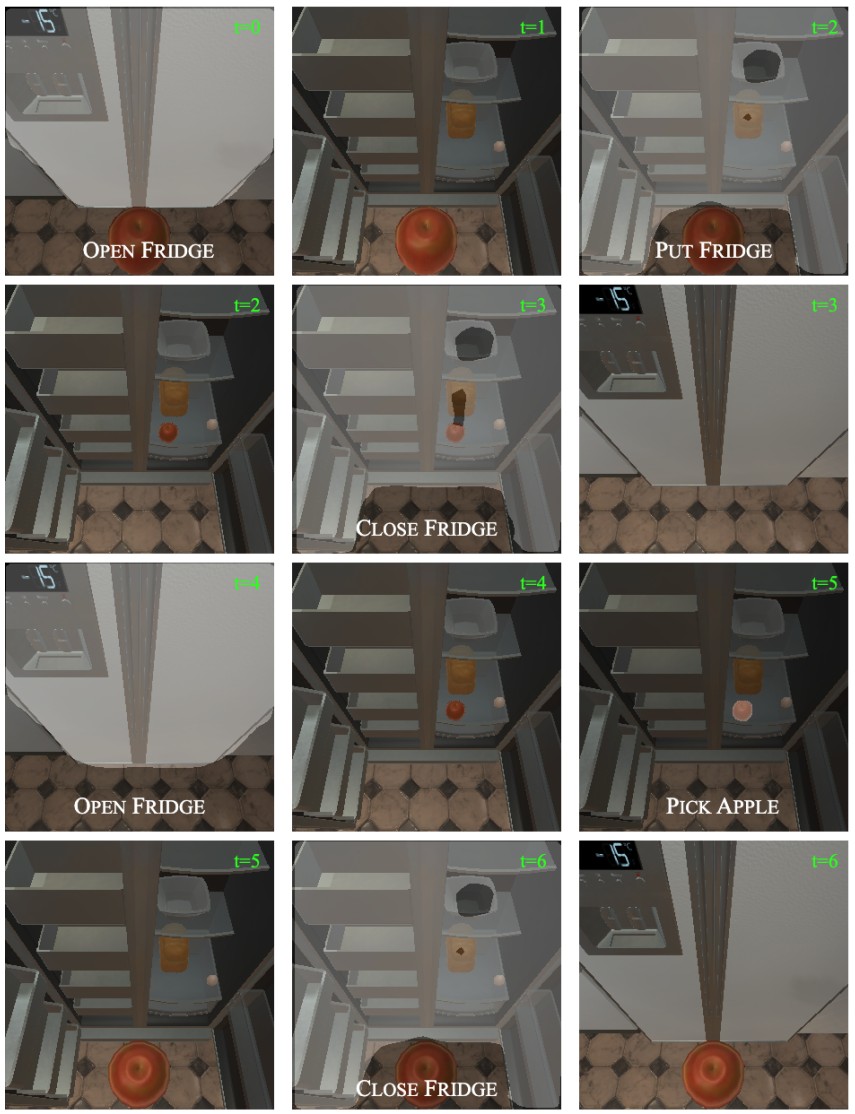

Figure 11: Cool Object interaction policy agent completes a subgoal task "Chill the apple in the refrigerator." in an unseen environment.

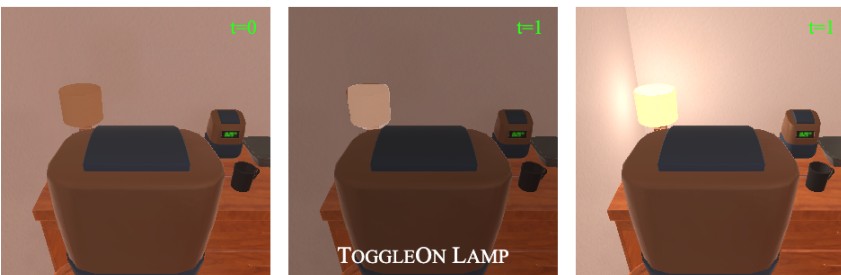

Figure 12: Toggle Object interaction policy agent completes a subgoal task "Turn on the lamp." in an unseen environment.

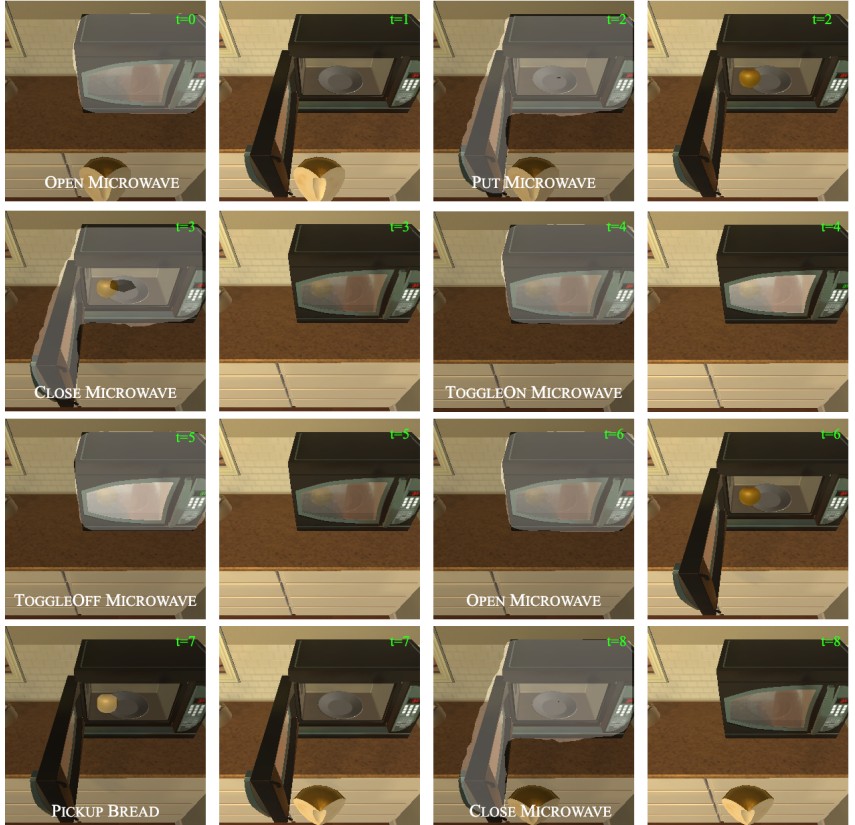

(a) Heat Object

Figure 13: Heat Object interaction policy agent completes a subgoal task "Heat the apple slice in the microwave." in an unseen environment.

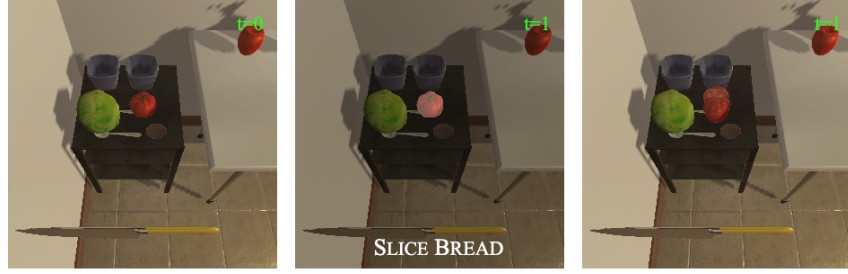

(a) Slice Object

Figure 14: Slice Object interaction policy agent completes a subgoal task "Make slices of the bread." in an unseen environment.

