# OpenReview forum: "Hierarchical Modular Framework for Long Horizon Instruction Following "
_ICLR.cc/2022/Conference — ICLR 2022 Submitted_

### Official Review · Reviewer_4rTh · 2021-10-19

**Correctness:** 4
**Technical Novelty And Significance:** 3
**Empirical Novelty And Significance:** 3
**Recommendation:** 8
**Confidence:** 4

**Main Review:**

Strengths:
1. The hierarchical and modularized approach seems promising for addressing the ALFRED problem, which is a mixture of navigation and multiple interactive actions. Decoupling and specifying different reasoning processes, as well as using independent state representations, greatly reduces the state space in learning, and as claimed in the paper, improves generalization ability and interpretability. I believe the method could benefit and influence lots of future work.
2. Comprehensive experiments in the main paper and in appendix nicely support the proposed methods and arguments in this paper. The performance of the agent on ALFRED has been improved by a significant margin.
3. The overall paper is nicely written, tables, figures are clearly presented.

Weaknesses / Comments:
1. A common limitation of learning specialized network for each subgoal is the increase in hardware cost, training cost and sometimes computational cost, especially when people want to design more complicated interaction policy or when there are a large number of subgoals. I am wondering if the authors can discuss about this point (and perhaps provide some quantitative analysis on this issue).
2. Following the previous comment, I believe the learning of some subtask can benefit each other, a simple example might be: heat and cool both requires the agent to put an object in to something, which such information is not shared across specialized policies. I would like to learn about the authors’ thoughts on this point.
3. There are several points which are unclear to me:
(1) Section 3.2.1, how to combine step-by-step instructions into a subtask instruction? I think ALFREAD only provide step-by-step instructions. I can understand this can be easily done in training because we know the ground-truth action correspond to each step-by-step instruction, but what about the validation data?
(2) Following previous point, it is unclear how does each step-by-step instruction x^hat_i translate to multiple subgoals. E.g. Figure 2 Instruction 3 contains navigation action and object interaction, but from Equation 1, it seems that each x^hat_i only translates into a single action. Please correct me if I misunderstood anything.
(3) The training of the seven interaction policies, e.g., training objectives. The paper also mentioned only two epochs of training is required, does it mean that the actions for completing each subgoal is similar (can be easily memorized? Or is it due to the limitation of the dataset?) I hope the authors can elaborate more on this point.


**Summary Of The Paper:**

This paper proposes a hierarchical and modularized framework for addressing the ALFRED language-guided task completion problem. The method divides instructions into navigational and interactive subgoals, where the navigation and each interaction is processed by a specialized policy. To further improve the agent’s performance, the paper introduces object encoding module and loop escape module. The proposed methods are novel and have been experimented comprehensively in this paper, which could benefit a range of future research.

**Summary Of The Review:**

The paper is nicely written, it presents interesting and valuable ideas, and the experiments are comprehensive. I am happy to accept the paper if the authors can clarify my questions.

---

> ### Author Response · Authors · 2021-11-21
> **Answers to the questions of Reviewer 4rTh (2/2)**
>
> > **Clarifications**
> > > **(1) Section 3.2.1, how to combine step-by-step instructions into a subtask instruction?** I think ALFRED only provides step-by-step instructions. I can understand this can be easily done in training because we know the ground-truth action corresponds to each step-by-step instruction, but what about the validation data?
>
> $\to$ In training, yes, we combine step-by-step instructions in the way you described. For validation data, the PCC predicts the subgoal for each step-by-step instruction at the beginning of the trajectory, and the instructions are combined using the PCC's output as the *pseudo* subgoal labels for the instructions.
>
>
> > > **(2) Following previous point, it is unclear how does each step-by-step instruction $\hat{x}_i$ translate to multiple subgoals.** E.g. Figure 2 Instruction 3 contains navigation action and object interaction, but from Equation 1, it seems that each $\hat{x}_i$ only translates into a single action. Please correct me if I misunderstood anything.
>
> $\to$ A single step-by-step instruction only translates to a single subgoal. We found the example in Fig. 2 confusing. In fact, the Figure 2’s instruction 3 (“Turn around, bring the potato to the microwave on the right”) corresponds to a single subgoal (*i.e.*, navigation to the microwave as **the agent already possesses the potato**). But instruction 3 may look like the combination of navigation and interaction as each word ‘Turn’ and ‘bring’ (*i.e.*, pick up) is for navigation and interaction, respectively. Thus, the word ‘bring’ together with ‘turn around’ only refers to the navigation. We change the example sentence in the Figure 2 to a less confusing one from the dataset (*’’Turn left and walk to the microwave‘’*) for clarity.
>
>
> > > **(3) The training of the seven interaction policies.** (a) Clarify e.g., training objectives.
>
> $\to$ We train the seven interaction policies in a supervised manner using behaviour cloning with cross-entropy losses between ground-truths and predictions for actions and object classes (*i.e.*, one for action and the other for object prediction for OCMP module). We add the details including a training objective, in Sec. A.1 in the revision.
>
>
> > > **(3) The training of the seven interaction policies.** (b) The paper also mentioned only two epochs of training is required, does it mean that the actions for completing each subgoal is similar (can be easily memorized? Or is it due to the limitation of the dataset?)
>
> $\to$ Yes, the action sequences corresponding to each subgoal are similar thus we only require two epoch training. The reason that the action sequences for a subgoal are similar is that a subgoal is associated with an object and the space of valid actions for an object is limited (*e.g.*, we do not ‘’Turn On’’ a ‘’knife’’). This reduces the space of possible action sequences and leads to small variations in the action sequences for a subgoal. We empirically observe that the success rate of the learned interaction policies do not improve significantly after two epochs of training (see Fig. 6 in the revision).

---

> > ### Comment · Reviewer_4rTh · 2021-11-24
> > **About Equation 6 in the revision**
> >
> > A negative sign missing for the loss function?

---

> > > ### Author Response · Authors · 2021-11-24
> > > **Thank you for the correction.**
> > >
> > > >**A negative sign missing for the loss function in Equation 6?**
> > >
> > > $\to$ Yes, you are correct. We will fix it in the final version. Thank you!

---

> > ### Comment · Reviewer_4rTh · 2021-11-25
> > **Execellent rebuttal and revision**
> >
> > Thanks the authors for the very comprehensive response to the reviews and the nice improvement on the paper. I will keep my score as 8 to accept the paper.

---

> > > ### Author Response · Authors · 2021-11-26
> > > **Thank you for valuable suggestions and questions**
> > >
> > > Thank you for your valuable suggestions and initiating an interesting discussion, as well as the encouraging comments, to improve the paper.

---

> ### Author Response · Authors · 2021-11-21
> **Answers to the questions of Reviewer 4rTh (1/2)**
>
> We appreciate encouraging remarks on the promising effectiveness of the approach, comprehensive experiments, strong results, and clear presentation. We address your detailed concerns as follows.
>
>
> > **A common limitation of learning specialized networks for each subgoal is the increase in hardware cost, training cost and sometimes computational cost.** Esp., more complicated interaction policy or a large number of subgoals. Discuss about this point (and perhaps provide some quantitative analysis on this issue).
>
> $\to$
>
> - **Hardware cost**: Our model would increase the memory cost linearly by the number of subgoals (0.23GB per subgoal) because models for each policy should be loaded in the memory proportional to the number of subgoals. Considering the large sized memory in modern computer systems (*e.g.*, usually larger than 8GB for PC and 6GB for high-end mobile phones), the additional memory cost may not be considered large, but it is certainly a cost.
>
> - **Training cost**: The training cost for each policy may marginally or may not increase because we divide a task into multiple subgoals and learning the specialized model for each subgoal takes a few epochs as the training set for each policy is *easier* (*i.e.*, sharing similar semantics) for training than the data for the *flat model*. In Figure 4 (a) in the revision, we show that learning the specialized networks requires less training epochs compared to learning a unified network. Therefore, even though there are multiple subgoals to be learned, the total training cost would increase marginally (and even embarrassingly parallelizable) or even decrease thanks to the reduced training cost per each subgoal added to an overhead of predicting subgoals.
>
> - **Computational cost**: We guess this is for inference (please correct us if we are wrong). It may marginally increase or even decrease because specialized networks for subgoals could infer each subgoal efficiently by reducing unnecessary exploration. Figure 4 (b) in the revision shows that the specialized networks require shorter mean episode lengths to accomplish tasks. With the efficient subgoal achievement, the overall computational cost at inference would be similar or even decrease, compared to the model with the flat policy learning.
>
>
> We add this discussion in Sec. A.8 of the revision.
>
>
> > **I believe the learning of some subtasks can benefit each other.** Heat and cool both require the agent to put an object into something. Sharing information across specialized policies would be helpful. Share authors’ thoughts on this.
>
> $\to$ This is a very interesting suggestion. We also believe that sharing information across subtasks would improve the learning efficiency. However, learning the commonalities across distinct subtasks would require a large amount of expensive annotated data with multiple tasks (actions) associated with a single object.

---

### Official Review · Reviewer_BjKc · 2021-10-25

**Correctness:** 3
**Technical Novelty And Significance:** 2
**Empirical Novelty And Significance:** 4
**Recommendation:** 8
**Confidence:** 4

**Main Review:**

## Short version
Overall, I think this paper presents an effective hierarchical approach to instruction following that’s tailored a bit too tightly to the ALFRED benchmark. I’m not too familiar with the VLN line of work, but I believe the authors have made a fair comparison to the existing works, based on the quantitative comparisons against baselines. Algorithmically, there seem to be a lot of moving parts that make the whole work well and achieve SOTA numbers, but can be interpreted as either clever, crafty or hacky, depending on how you look at it. I respect the effort put by the authors in carefully designing each component and engineering an effective algorithm for the ALFRED benchmark but have concerns regarding the generality of the findings.

---
## Strengths
1. The writing and presentation is very effective and the authors do a great job of explaining the components of the algorithm in an intuitive manner.
2. The ablations, although a bit binary, are exhaustive and do a great job of highlighting the importance of each component in isolation. The ablations are conducted like a well-written CVPR/ICCV paper, which tend to have some of the most thorough ablation experiments.
3. The task that the paper addresses is extremely visual and the authors do a good job of visually showing rollouts and their method in action. It really highlights the long-horizon and semantic nature of the problem and intuitively describes the method (fig 1, 5).
4. The implementation and experimental details, while not exhaustive, feel sufficient to understand and replicate the settings in which the results are presented. The impact of the paper will be greatly improved by a public code release.

---
## Weaknesses/Concerns

1. My biggest concern with the method is that it seems overfit to the ALFRED benchmark and a lot of the design decisions were motivated by common failure cases in the benchmark. While that can be a great way to make progress on a challenging task like embodied instruction following, I’m not convinced that the algorithm and insights necessarily generalize to the broader range of problems. Specifically, the “loop escape” component or the “OEM module”, while solving real problems, feel a bit too specific and poorly justified. This is also depicted in how these components are motivated — e.g. “”To further improve the task completion performance, we propose…”.
2. The authors make several claims that seem poorly motivated or not justified:
	(i) [Section 3] “navigation and interaction must be distinguished because the former requires a broader set of information” — this could benefit by a proper discussion or references to prior work. While it’s true that almost all prior/concurrent work deals them independently, it’s almost a _bug_ and not a _feature_; we could use systems that better integrate these two capabilities. I am not saying that the choice of treating them independent is a wrong one (it does make things easy), but the claim needs to be qualified/justified a bit.
	(ii) [Section 3.2] “this necessitates an independent module that can…” — why is this the case? This feels more like a by-product and not requirement.
	(iii) [Section 4.3] “… highlights the modular structure’s benefits in long-horizon planning and superior generalization capabilities” — how does it do so? It’s not clear to me how the single policy metrics imply this. A bit more clarification would help here, since “better generalization” is presented as a key advantage of the proposed method at the start.

3. [Section 3.2.4] Regd. the loop escape module: $W$ being a hyperparameter feels a bit crafty. The job of the “loop escape” is to break out of a deadlock, and being able to do so for a fixed/preset value of repetition (which must be set for “best results”) again sounds like something that is overfit to the benchmark. In a general set of environments, this value could vary drastically and there could be a better way to handle this (if at all such a module is necessary).

4. [Section 3.1] Regd. the policy composition controller: I get the impression that the benefits of this are overstated, and the main purpose of this controller is to learn correspondence between “semantic subgoals” and text instructions. According to Eqn. 1, these subgoals seem to be predefined, and hence, the mapping is pretty straightforward when done in a supervised manner and isolated for the controller. The claims about interpretability need to be toned down in this case.

5. [Section 3.1] I had another question regarding the PCC: Based on the description, it seems to me that the PCC learns a one-to-one mapping from $\hat{x}_i$ to subgoals $s_i$. Is that correct? If so, how does this handle instructions that need multiple subgoals, e.g. “heat the potato in the microwave” would need HeatObject and GoTo etc. Either way, a bit more clarification on how the instructions are handled could be useful in either the main text or appendix.

6. While this may not be an outright weakness, a lot of the proposed contributions lead the work towards a “classical” pipeline, with individual stages for parsing inputs, perception/object segmentation, running control, loop detection/closure etc. Looking into the progress made in embodied AI in the robotics and vision communities, a lot of these _hand-engineered_ stages have benefitted from being replaced by data-driven means and going back to such a pipeline is intriguing —  it brings up the age-old question about what to learn v/s what to bake in as an inductive bias. While more instrumentation like the proposed method seems to improve benchmark metrics, it might come at the cost of “generalization of insights”. This relates to the earlier point (1), and a discussion about this philosophy would be nice. (I don’t hold this against the paper and this does not factor in my assessment, just a useful discussion to include.)


---
## Misc. Comments/Typos
I noticed a couple of typos in the text that you can fix. Don’t worry, this does not affect my impression of the otherwise well-written paper :)

- [Section 1] “To evaluate our proposal in a challenging scenario*s*”
- [Figure 2, caption] “comprises of *tree* main components”


---

*Update*: Updating my recommendation to reflect the discussion by the others.

**Summary Of The Paper:**

The paper looks at the problem of instruction following (using language) in a navigation+interaction task setting, encapsulated by the ALFRED benchmark. The authors propose a hierarchical modular approach that operates on three levels: (i) identifying subgoal sequence, (ii) a navigation “master” policy, and (iii) a manipulation/“interaction” policy. Evaluations against a slew of baselines from the vision+language navigation literature are presented, showing that the proposed method (HACR) outperforms existing baselines in both seen and unseen tasks. Careful ablations are also presented.

**Summary Of The Review:**

The paper presents an effective algorithm for instruction following within the ALFRED benchmark but the means and insights seem overfit to the metrics. I am on the fence regarding acceptance and look forward to a response from the authors.

---

> ### Author Response · Authors · 2021-11-22
> **Answers to the questions of Reviewer BjKc (3/3)**
>
> > **[Not an outright weakness] A lot of the proposed contributions lead the work towards a “classical” pipeline.** It is with individual stages for parsing inputs, perception/object segmentation, running control, loop detection/closure etc. It is intriguing that it brings up the age-old question about what to learn v/s what to bake in as an inductive bias.
>
> $\to$ We thank the reviewer for initiating this very interesting discussion and are happy to share our thoughts on it. First, we agree that our method is a more or less ‘classical’ pipelined approach and has many carefully designed components as a set of inductive biases. We attribute our design to (1) lack of sufficient training data and (2) requirement of heavy computational cost. (1) Given our framework is the behavior cloning (BC) [F], though the benchmark we used is the largest in the literature, the rigid nature of the BC approach (*i.e.*, any deviation from the expert trajectory over time is not allowed to learn a behavior) requires vast amount of training examples (*i.e.*, may be not sufficient to be the largest dataset in the literature) to learn a *satisfactory* model. (2) To overcome the rigidity of learning a BC model, there are many approaches proposed in the literature [G,H]. But they are mostly computationally expensive [I]. We aim the niche between both problems by designing well crafted components for better encoding the given dataset even in the rigid BC framework.
>
> Reference
> [F] M. Bain & C. Sommut. A framework for behavioural cloning. In Machine Intelligence, 1999.
> [G] J. Ho & S. Ermon. Generative adversarial imitation learning. In NeurIPS, pp. 4565–4573, 2016.
> [H] J. Fu et al. Learning robust rewards with adversarial inverse reinforcement learning. In ICLR, 2018.
> [I] I. Kostrikov et al. Discriminator-actor-critic: Addressing sample inefficiency and reward bias in adversarial imitation learning. In ICLR, 2019.
>
>
> > > While more instrumentation like the proposed method seems to improve benchmark metrics, it might come at the cost of “generalization of insights”. This relates to the earlier point (1), and a discussion about this philosophy would be nice. (I don’t hold this against the paper and this does not factor in my assessment, just a useful discussion to include.)
>
> $\to$ Similar to our answer to the previous question, our inductive bias could be useful for this task not just for this benchmark by the analogy of convolutional neural networks for images. By the universal approximation theorem, the 2D information, *e.g.,* images, could be encoded by a multi-layer perceptron (MLP) with sufficient amount of data. Given that it is difficult to fathom the sufficient amount of data to train the MLP, we have a very successful design choice for encoding 2D information; the patch wise encoding and convolutional operations for its processing. Now, with 1M images, *e.g.*, the ImageNet-1K, we can have a decently performing image classifier. We hope that our designed components could be useful to inspire the future research in designing a data driven model to address the Embodied AI task successfully. In sum, we pose our work as a stepping stone for designing and learning a successful data driven model in the future. We add this discussion in Sec. A.7 of the revision.
>
>
> > Misc. Comments/Typos.
>
> $\to$ Thank you for the corrections. We revise them in the revision.

---

> ### Author Response · Authors · 2021-11-22
> **Answers to the questions of Reviewer BjKc (2/3)**
>
> > **Motivation and justification for claims**
> > > **(ii) [Section 3.2]** “this necessitates an independent module that can…” — why is this the case? This feels more like a by-product and not a requirement.
>
> $\to$ Yes, it is a design choice but not a necessity. We revise the misleading sentence in the revision. Thank you!
>
>
> > **Motivation and justification for claims**
> > > **(iii) [Section 4.3]** “… highlights the modular structure’s benefits in long-horizon planning and superior generalization capabilities.” How does it so? It’s not clear to me how the single policy metrics imply this. Need clarification about modular structure’s benefits for better generalization.
>
> $\to$ We found that this analysis is wrong since the modular structure can learn the environment in training well but not very well generalizable to unseen environments (it is partly attributed to the imitation learning framework with insufficient data -- also refer to our answer to your discussion request on ‘pipelined approach’ below). We apologize for the confusion. We remove this from Sec. 4.3 of the revision.
>
>
> > **Motivation and justification for claims**
> > > **[Section 3.2.4] Regd. the loop escape module: $W$ being a hyperparameter feels a bit crafty.** Value of $W$ is tuned for benchmark and in a general set of environments, this value could vary drastically and there could be a better way to handle this (if at all such a module is necessary).
>
>
> $\to$ We obtain $W=10$ by a grid search (from 0 to 25) on the validation set. We observe that variation over different values is not large (up to 0.36%), implying that the performance is not very sensitive to this hyperparameter. We add this sensitivity analysis in Figure 5 in Sec. A.6 in revision.
>
> For an idea to better handle this, we could consider a learning-based approach to escape deadlock states (Du et al., 2020) but it requires interaction with a simulator for deadlock experience. Since the benchmark we use for the experiments is an offline dataset (*i.e.*, a simulator is not used during training), variability in visual information during training is limited. Thus, it is non trivial to use a learning based approach for this task.
>
>
> > **[Section 3.1] Regd. the policy composition controller**: Benefits of this seem overstated. Main purpose of this controller is to learn correspondence between “semantic subgoals” and text instructions. According to Eqn. 1, these subgoals seem to be predefined, and hence, the mapping is pretty straightforward when done in a supervised manner and isolated for the controller. The claims about interpretability need to be toned down in this case.
>
> $\to$ We guess that the reviewer refers to the claim of ‘The PCC’s predictions correlate to semantic subgoals, … in a more interpretable way than the low-level actions.’ in Sec. 3.1. We also found that the benefits of the Policy Composition Controller (PCC) are overstated as the instruction to subgoal mapping is straightforward. We tone down this argument to “The PCC’s predictions correlate to semantic subgoals, ... which enables us to monitor the progress of task completion by the agent.” in the first paragraph in Sec. 3.1 of the revision. If we misunderstood the part we overstated, please let us know. Thank you.
>
>
> > **[Section 3.1] I had another question regarding the PCC**:  Based on the description, it seems to me that the PCC learns a one-to-one mapping from $\hat{x}_i$ to subgoals $s_i$. Is that correct?
>
> $\to$ Yes, you are correct.
>
>
> > > If so, how does this handle instructions that need multiple subgoals, e.g. “heat the potato in the microwave” would need HeatObject and GoTo etc. Either way, a bit more clarification on how the instructions are handled could be useful in either the main text or appendix.
>
> $\to$ Currently, PCC does not handle instructions that correspond to multiple subgoals (as our answer to the previous question). The example instruction 4 (“heat the potato in the microwave”) in figure 2 is confusing; it refers to a single subgoal ( *i.e.*, HeatObject) only, where the agent is directed to heat the potato, assuming that the agent is *already in front of* the microwave by previous action sequences. We revise the example sentence in figure 2 to a less confusing one from the dataset (“Heat the potato in the microwave”) in the revision. Apologies for the confusion.
>
> To handle multiple-subgoal instructions, we may use a sequential model as the PCC to generate multiple subgoals for an instruction (*i.e.*, one-to-many mapping) in a supervised manner by the help of a large annotated data. We add this clarification on how instructions are handled in Sec. 3.1 in the revision.

---

> ### Author Response · Authors · 2021-11-22
> **Answers to the questions of Reviewer BjKc (1/3)**
>
> We appreciate encouraging remarks on intuitive presentation and thorough ablations. We address your detailed concerns as follows.
>
>
> > **Overfit to the ALFRED benchmark.** A lot of the design decisions were motivated by common failure cases in the benchmark. Not convinced that the algorithm and insights necessarily generalize to the broader range of problems. Specifically, the “loop escape” component or the “OEM module”, while solving real problems, feel a bit too specific and poorly justified.
>
> $\to$ We respectfully argue that these components are not specifically designed for the benchmark but for the task of instruction following. Specifically, the two modules are for navigation, which is one of the challenges of the instruction following task. We detail the motivation for each module as follows.
>
> - **Loop escape**: Inspired by human trial-and-error behavior, we circumvent the deadlock states by modeling the ‘detecting locking in the loop’ as *trial* and ‘a rule based inductive bias’ as a way to escape from *error*. Similar technique has been used for visual navigation method in Embodied AI literature (Du et al., 2020). In addition, it only requires RGB observations on which many benchmarks [Anderson et al., 2018; Krantz et al., 2020; A, B, C] rely. Therefore, it could be used for any tasks that require navigation with visual observation to discover and escape deadlock states, not only for the benchmark we used.
>
> - **OEM module**: It predicts information about an object from an instruction for navigation. Unlike ‘object oriented navigation’ that uses given object information as input (Yang et al., 2018; Wortsman et al., 2019), we address a realistic instruction following setup that needs to *infer objects from language instruction (that may contain multiple object words) for navigation* by the proposed OEM. Therefore, this module can be useful for other realistic language based Embodied AI tasks such as vision-and-language navigation (Anderson et al., 2018; Krantz et al., 2020), visual referring expression [D], and *etc.*
>
>
> We add these motivational arguments for each component in the 4th paragraph in Sec. 1 of the revision.
>
>
> References:
> [A] L. Weihs et al. Visual Room Rearrangement. In CVPR, 2021.
> [B] K. Ehsani et al. ManipulaTHOR: A Framework for Visual Object Manipulation. In CVPR, 2021.
> [C] K.-H. Zeng et al. Pushing it out of the Way: Interactive Visual Navigation. In CVPR, 2021.
> [D] Y. Qi et al. REVERIE: Remote Embodied Visual Referring Expression in Real Indoor Environments. In CVPR, 2020.
>
>
> > **Motivation and justification for claims**
> > > (i) [Section 3] “navigation and interaction must be distinguished because the former requires a broader set of information” — this could benefit by a proper discussion or references to prior work.
> > > > While it’s true that almost all prior/concurrent work deals with them independently, it’s almost a *bug* and not a *feature*; we could use systems that better integrate these two capabilities. I am not saying that the choice of treating them independently is a wrong one (it does make things easy), but the claim needs to be qualified/justified a bit.
>
> $\to$ We also found that the claim is not well justified (*e.g.*, must be distinguished) since we agree that it is a better integrated model in the future may outperform the hierarchical models. Nonetheless, we want to discuss our intuition for that claim as follows:
>
> - The class imbalance between atomic actions for navigation and interaction (*i.e.*, navigation actions are far more frequent than the interaction actions) would bias a model towards more frequent navigation actions (*i.e.*, navigation). Following a large body of literature that multiple models help address the class imbalance [E], we propose to use a separate model for each action.
>
> - The visual observation for navigation varies considerably over time while for object interaction it is largely stationary. Therefore, the navigation policy needs to learn the global environment in a relatively long time horizon, whereas the manipulation policy requires focusing on local visual cues and spatial relationships among objects in the current visual observation for mask generation. As one is visually dynamic in time whereas the other is visually less diverse but requires detailed reasoning, encoding both by different architecture may be effective.
>
>
> We add these arguments for our claim in the first paragraph in Sec. 3 of the revision.
>
> References:
> [E] M. Galar et al., "A Review on Ensembles for the Class Imbalance Problem: Bagging-, Boosting-, and Hybrid-Based Approaches," IEEE Trans. on Systems, Man, and Cybernetics, Part C (Applications and Reviews), vol. 42, no. 4, pp. 463-484, July 2012

---

> > ### Comment · Reviewer_BjKc · 2021-11-25
> > **Excellent rebuttal, bumping up score!**
> >
> > Thank you for the extremely detailed, thoughtful response and your candor in the discussions. I really appreciate you acknowledging the limitations of the claims (and consequently modifying the paper).
> >
> > I have updated my score to reflect the updated manuscript and the authors' responses.

---

> > > ### Author Response · Authors · 2021-11-26
> > > **Thank you for valuable suggestions and clarification questions**
> > >
> > > We appreciate your valuable suggestions with the encouraging remarks and clarification requests for unclear claims/underlying intuitions to improve the paper. Thank you!

---

### Official Review · Reviewer_KXWq · 2021-11-02

**Correctness:** 3
**Technical Novelty And Significance:** 2
**Empirical Novelty And Significance:** 3
**Recommendation:** 8
**Confidence:** 4

**Main Review:**

**Strength**

- The work attacks a challenging problem
- The idea is appealing and intuitive: making the  decomposing instructions into hierarchical subgoals and use different policy modules to handle them
- Good empirical performance on competitive benchmark

**Weaknesses**

- Aside from the main claims, the paper lacks some critical details (see question 1) and is a bit confusing (see questions 2-4). See the questions below for concrete examples.
- Hard to really pinpoint what the main contribution is. There are claims such as “novel language instructions” in sec 3.2.1 or “we propose a module” in sec 3.2.2. The above two examples are not mentioned in the main claims but they obscure the main points as I’m not sure if they are technical details or a contribution. Similar issue in the ablation study. The paper ablated on aspects that are not mentioned in the main claims such as DA, NIH, and OCMP. It’s unclear what conclusion the readers should draw from the analysis.
- Lacks analysis compared with previous best models. The performance improvement seems large, but the paper provides little insight in terms of what gains it gets against similar hierarchical approaches (Zhang and Chai, 2021, Blukis et al., 2021). E.g., comparison in Table 2 is too weak (flat policy is a weak baseline); should compare to similar hierarchical approaches referenced above.
- No significance tests

**Questions**
- Q1: There is prior work attempting to make instruction following hierarchical, which is duly cited in Related Work (e.g., Blukis et al., 2021). But I couldn’t parse the fundamental difference between “3 layer hierarchy” v.s. “1 layer hierarchy” in prior work. As far as I understand, prior work does generate subgoals which group low-level actions. If it’s pointing at the 3 layer modular structure, then I’d like to understand how having architectural hierarchy is useful in addition to action hierarchy
- Q2: How is encoding the instructions with Bi-LSTM a novel approach? (sec 3.2.1)
- Q3: Where does the subgoal space come from? Is this designed by the authors?
- Q4: Why does <manipulate> appear again in the master policy? Shouldn’t this be for the PCC to decide?


**Summary Of The Paper:**

The paper proposes a hierarchical approach (HACR) to tackle the long-horizon issue for embodied instruction following tasks. The hierarchical structure includes
- a Policy Composition Controller (PCC): predicts a sequence of predefined subgoals
- a Master Policy: decides whether to manipulate or keeps navigating
- an Interactive Policy: performs the low-level interaction.

The main contribution claimed in the paper is the decomposition of the high-level goal into subgoals (which makes it interpretable), adding OEM and LEM to boost performance, and the SoTA performance on ALFRED benchmark.

**Summary Of The Review:**

Overall, I think it’s a worthwhile paper if the empirical result is correct. The main weakness lies in the structure and the presentation of the paper which make it 1) hard to see the main contributions (is the main claim that hierarchical > flat or with the combination of the proposed components, MIP, OCMP, FIP, etc it works better than other hierarchical approaches?) 2) hard to draw insight as its comparison against prior best models is limited (e.g., does HACR also converge faster compared with other hierarchical approaches in Fig 4?). I would be happy to raise the scores if the authors could clarify more on the above two points as well as answering my questions.

---

> ### Author Response · Authors · 2021-11-22
> **Answers to the questions of Reviewer KXWq (3/3)**
>
> > **Significance tests.**
>
> $\to$ For the significance tests, we repeat the experiments 5 times (with different seed) for our full model and the ablated ones, and summarize the average performances with their standard deviation in Table 2 (MIP, NIH, and OEM components) and Table 3 (OCMP, FIP, and DA components).
>
> As shown in the full model’s standard deviations (0.2), our model exhibits marginal deviations in all metrics. Compared to the other methods in Table 1, with small deviation of performance, we could claim that our method is statistically significant (unfortunately, other models did not report the standard deviations).

---

> > ### Comment · Reviewer_KXWq · 2021-11-28
> > **Updated the scores**
> >
> > I want to thank the authors for the very clear and detailed response. As stated in the original review, I think the paper brings value to the community by showing that suitable modular + hierarchical design can lead to significant performance boost on a commonly recognized benchmark. I observe the concerns from other reviewers on how representative the benchmark is and the risk of overfitting to the task. I agree that it's an important discussion for the community. Nonetheless, I think it's orthogonal to this particular paper's value. I am happy to raise the scores based on that my confusions are addressed and the contributions are now much clearer. Thanks again for the effort.
> >
> > -Reviewer KXWq

---

> ### Author Response · Authors · 2021-11-22
> **Answers to the questions of Reviewer KXWq (2/3)**
>
> > **W2: Main contribution**
> > > **1. Two examples are not mentioned in the main claim and obscure the main point.**; “novel language instructions” in sec 3.2.1, “we propose a module” in sec 3.2.2. Are they technical details or a contribution?
>
> $\to$ The main contribution is a combination of novel and existing components (see our answer to the question right below for details). These two are technical details for the two novel components as follows.
> - **‘language instruction combination’**: it is a different way of encoding the instructions for navigation and manipulation (see our answer to your question Q2 for detail).
> - **‘object encoding module’**: it acts as a navigation subgoal monitor, indicating if the agent has found the object for interaction. It is a simple yet effective method for improving the navigation precision by providing a stopping criterion for the agent at inference time which has not been discussed in the literature.
>
>
> > > **2. Similar issue (obscurity of main point) in the ablation study.** The paper ablated on aspects that are not mentioned in the main claims such as DA, NIH, and OCMP. It’s unclear what conclusion the readers should draw from the analysis.
>
> > > **In Summary of the Review:** Is the main claim that hierarchical > flat or with the combination of the proposed components, MIP, OCMP, FIP, etc works better than other hierarchical approaches?
>
> $\to$ Our main contribution is the combination of navigation interaction hierarchy (NIH), modular interaction policy (MIP), and object encoding module (OEM). To showcase the benefits of the combined framework, we provide ablation of each of these components in Table 2 in revision. We observed that ablating any of the components degrades the agent’s task performance by noticeable margins. We clarify this in Sec. 4.4 in revision.
>
>
> Please note that Table 2 in the original submission presents the ablation study on model’s components (OCMP, FIP, and DA) and the main contributions (NIH, MIP, and OEM). For clarity, we divide Table 2 in the original submission into two tables in the revision; Table 2 for the main contributions and Table 3 for the design components. We are sorry about this confusion and appreciate the comments.
>
>
> > **W3: Analysis compared with previous best models.**
> > > **(a) Comparison in Table 2 is too weak (flat policy is a weak baseline).** Should compare to similar hierarchical approaches referenced above.
> > > **In summary of the review** Limited comparison against prior best models.
>
> $\to$ By the suggestion, we compare the prior hierarchical works (Zhang & Chai, 2021; Blukis et al., 2021) with our approach in Table 5 in appendix in the revision. Compared to the similar hierarchical approaches, we observe that HACR achieves better performance in most task types (5/7).
>
>
> > > **(b) Analysis compared with previous best models.** Provide insights of what gains it gets against similar hierarchical approaches (Zhang & Chai, 2021, Blukis et al., 2021);
>
> $\to$ In our **3-layer hierarchy**, we process navigation and multiple interactions using individual modules whereas in **prior hierarchical works** (Zhang et al., 2021; Blukis et al., 2021), they process the navigation and interaction subgoals with a single low-level policy. With the architectural hierarchy along with action hierarchy, the agent learns navigation and interaction in separate modules. Our intuition about the necessity of the separation is that it allows each module to effectively handle the class imbalance between low-level actions for navigation and interaction subgoals (since navigation trajectories have far more actions than the interaction ones). In addition, it helps in learning different degrees of variations in visual observations for navigation and interaction separately. Our empirical results support our design (compare the first row and row (b) in Table 2 in revision).
>
> Additional benefits of using the 3-layer hierarchy include *easy inclusion of additional interaction subgoals* than the other works, because we only need to train the newly added interaction module on the expert trajectories (short-horizon) unlike others.
>
>
> > > **In summary of the review** Does HACR also converge faster than other hierarchical approaches?
>
> $\to$ We are not sure. We have no information about the convergence of prior hierarchical approaches to make a conclusive remark. *E.g.*, (Zhang & Chai, 2021) mentions they train their network for 10 epochs (less than ours) but they utilise a pretrained RoBERTa model whose training cost is not discussed. Similarly for (Blukis et al., 2021), the github repository (made public after submission deadline) shows the number of training epochs for different modules which indicate that it converges faster than our approach in terms of low-level policy learning. But we are eager to compare our method with HACR in computational cost.

---

> ### Author Response · Authors · 2021-11-22
> **Answers to the questions of Reviewer KXWq (1/3)**
>
> We appreciate encouraging remarks on attacking a challenging problem, appealing and intuitive ideas, and good empirical performance. We address your detailed concerns as follows.
>
>
> > **W1: Lack details: Q1-A: Fundamental difference between “3 layer hierarchy” v.s. “1 layer hierarchy” in prior work (Blukis et al., 2021)?** As far as I understand, prior work does generate subgoals which group low- level actions.
>
> $\to$ In **1-layer hierarchy** (actually it is a 2-layer hierarchy (revised in the revision). Apologies for the confusion), yes, it generates subgoals which group low-level actions and they process the interaction subgoals by a single network. In contrast, in our **3-layer hierarchy**, we process the high level subgoal planning, navigation actions and interaction actions using three independent modules in a hierarchical relation.
>
>
> > **W1: Lack details: Q1-B: If the fundamental difference is pointing at the 3 layer modular structure, I’d like to understand how having architectural hierarchy is useful in addition to action hierarchy**
>
> $\to$ Yes, the **3-layer hierarchy** points at the 3 layer modular structure. We guess that for the ‘architectural hierarchy’, the reviewer refers to using separate networks for subgoal planning, navigation actions, and interaction actions (please correct us if we are wrong). With the architectural hierarchy along with action hierarchy, the agent learns navigation and interaction in separate modules. Our intuition about the necessity of the separation is that it allows each module to address the class imbalance between low-level actions for navigation and interaction subgoals. This is because navigation trajectories have far more actions than the interaction ones. In addition, it helps in learning different degrees of variations in visual observations for navigation and interaction separately. Empirically, this leads to better performance.
>
>
> > **W1: Confusing: Q2: How is encoding the instructions with Bi-LSTM a novel approach? (sec 3.2.1)**
>
> $\to$ We found that the way to combine language instructions for navigation is not properly stated. We clarify this in Fig.2, its caption and Sec. 3.2.1. **For a navigation action (addressed by the master policy)**, both the current time step’s navigation action and the next time step’s manipulation action are concatenated to be fed to the Bi-LSTM (we revise ‘navigation policy’ in the MP block of the Fig.2 accordingly) along with object information ($o_t$) and previous time step’s action ($a_{t-1,n}). Our intuition for this is that the navigation with a *purpose of doing the next action* with the knowledge of object to interact helps in efficient navigation (*e.g.*, for the instructions (1) *Turn around and walk forward to the garbage bin.* and (2) *Pick up the blue credit card on the TV stand.*, the agent needs to locate and interact with the credit card, which is important information for agent to *know when to stop* navigation, *i.e.*, the agent should stop if it encounters the credit card in close vicinity.). In contrast, for a **manipulation action (addressed by the interaction policy)**, only the current time step’s manipulation instruction is used as the input to the policy. We claim that the scheme to use different instruction encodings for either navigation or manipulation is novel (the Sec.3.2.1 is presenting an example of it).
>
>
> As a side note, this proposed scheme improves navigation empirically by the provided information about the target object; using only the ‘navigation instruction’ degrades the agent’s performance as shown in row 6 of Table 4 in revision (Table 3 in the original submission). Please note that most of the prior work either concatenate all the step-by-step instructions (Shridhar et al., 2020; Singh et al., 2021; Pashevich et al., 2021; Kim et al., 2021; Zhang & Chai, 2021) or process them separately (Nguyen et al., 2021).
>
>
> > **W1: Confusing: Q3: Where does the subgoal space come from?** Is this designed by the authors?
>
> $\to$ We use the seven subgoals defined in the used benchmark. Note that these subgoals commonly appear in a variety of embodied AI problems such as object interaction (Zhu et al., 2017), rearrangement [A], instruction following (Shridhar et al., 2020), and *etc.*. We add the details in Sec. A.3 (paragraph 1) in the revision.
>
>
> References:
> [A] L. Weihs et al. Visual Room Rearrangement. In CVPR, 2021.
>
>
> > **W1: Confusing: Q4: Why does <manipulate> appear again in the master policy?** Shouldn’t <manipulate> in MP be for the PCC to decide?
>
> $\to$ Apology for the confusion. The ‘manipulate’ in the MP is different from that in the PCC. ‘Manipulate’ in the MP is an *action token* that indicates the *end of navigation* whereas ‘Manipulate’ in PCC determines whether the subgoal is a manipulation subgoal or not. We found this confusing. So, we change the notation of ‘Manipulate’ in PCC to ‘GotoLocation?’ in Figure 2 in the revision for clarity.

---

### Official Review · Reviewer_pj1s · 2021-11-02

**Correctness:** 3
**Technical Novelty And Significance:** 2
**Empirical Novelty And Significance:** 2
**Recommendation:** 3
**Confidence:** 4

**Main Review:**

**2. Strengths**

The idea of hierarchical strategy sounds technically correct and the ablation study in the experimental section shows its better performance than the previous method. The method also provides more transparent execution processing since the strategy predicts the subgoal during inference besides the step-by-step instruction.

**3. Concerns**

The reviewer has many concerns about the method which the reviewer thinks resolving them will make it a better paper.

**(1). About the module design.**

The PCC and MP modules do not make any use of the visual input for subgoal prediction and navigation action prediction. This looks strange since it will always be easier for humans to see the room before making subgoal and action predictions. It will be better to verify visual input's effectiveness with experiments. At least, the reviewer thinks that a discussion of the usage of visual input for PCC and modules are needed.

**(2). About the fairness of the comparison.**

The method uses additional supervision signals (the subgoal labels) compared with Singh et al (ICCV 2021). Also, it adopts a low-level text instruction as input when compared with HLSM. It's OK to use these additional signals as input but the reviewer thinks that it will be better to clarify them in the comparison.

**(3). About performance comparison.**

The performance gap in the table between HLSM and HACR is small (16.70 to 16.29). We care more about the unseen testing split and so does the official Alfred challenge. The authors of HLSM recently release their code publicly on github (https://github.com/valtsblukis/hlsm.git) and achieve much better performance (20.27) with better parameters fine-tuning. It is OK not to compare with the updated performance in this submission since the code has not been released at that time. However, a discussion about how HACR is better and complementary to HLSM will be more convincing. Note that HLSM uses less supervision signals.

**(4). About the overfitting issues.**

The performance of HACR seems to suffer from strong overfitting in table 1 (from seen to unseen and from validation to test). More analysis for such a performance gap is needed and makes the paper stronger.

**Summary Of The Paper:**

**1. The main idea of the paper**

This paper introduces a hierarchical strategy for interactive instruction following. Following previous module networks[Singh et al., ICCV 2021 and Corona et al. 2020] for interactive instruction learning, the author adopts module networks for action prediction. Different from the previous module networks, the authors propose to learn the middle-level goals first and decompose the step-by-step action predictions into two modules, master policy for navigation and interaction policy for object interaction. To better improve the performance, the authors also provide some minor modules like the object encoding module and the loop escape module. The method overall achieves slightly better performance than the previous method.



**Summary Of The Review:**

According to the concerns in the **Main Review**, the author currently tends to reject the paper. The viewer also believes addressing the issues will make the paper stronger.

---

> ### Author Response · Authors · 2021-11-21
> **Answers to the questions of Reviewer pj1s**
>
> Thank you for encouraging remarks on better performance and a transparent approach. We address your concerns as follows.
>
>
> > **Module design.** PCC and MP do not use visual input but it will always be easier for humans to make subgoal and action predictions with visual input. Verify visual input's effectiveness with experiments for PCC and MP.
>
> $\to$ We use visual input in MP (Please refer to the green circle to ‘Navigation Policy’ in MP. Apology for the unclear subtle presentation) but not in PCC. Our intuition for not using visual input for PCC is that the language instructions would contain sufficient information for high-level subgoal planning and the visual input might contain unnecessary visual details that do not help planning. As suggested, we conduct an ablation study for PCC and MP with and without visual input in Table 7 and 8 in Sec. A.6 in the revision. The results empirically support our choice as follows:
>
> - In Table 7, PCC with the visual input achieves lower success rates in both seen and unseen splits. We believe that this happens due to the ‘causal misidentification’ phenomenon [A]; when receiving *more* information (*e.g.*, RGB observation), the agent could learn to exploit irrelevant information (*e.g.*, a chair in the room) to a target task (*e.g.*, Heat a potato) rather than necessary information (*e.g.*, Microwave and potato), leading to performance drop.
>
> - In Table 8, in contrast, visual input for MP helps in the success rate.
>
> References:
> [A] P. de Haan et al. “Causal confusion in imitation learning.” In NeurIPS, 2019.
>
>
> > **Fairness of the comparison.** HACR uses additional supervision, compared with Singh et al (ICCV 2021). It adopts a low-level text instruction as input when compared with HLSM. It's OK to use these additional signals as input but it will be **better to clarify the additional supervision in the comparison**.
>
> $\to$ Clarifying the supervision for each method is a great suggestion! We add a column for the supervision that each method uses in Table 1 in the revision. As the reviewer may imply (by *it’s OK to use additional signals as input*), each supervision has pros and cons and it is not trivial to determine which supervision is stronger than others; *e.g.*, HLSM uses depth supervision to estimate the 3D layout of the environment for rich visual understanding while our HACR exploits language instructions for action planning with the given subgoal labels by the benchmark (*i.e.*, we do not make additional efforts to collect it). Note that other methods compared in the table (*e.g.*, Zhang & Chai, 2021; Blukis et al., 2021) use different sets of supervision for winning the benchmark.
>
>
> > **Performance comparison.** The performance gap in the table between HLSM and HACR is small (16.70 to 16.29). The challenge cares more about unseen testing split. A recently released code of HLSM achieves much better performance (20.27). It is OK not to compare with the updated performance in this submission. But discuss how HACR is better and complementary to HLSM. Note that HLSM uses less supervision signals.
>
> $\to$ We believe that HLSM and HACR are complementary because HLSM uses depth supervision to estimate the 3D layout of the environment for detailed visual understanding while our HACR exploits high-level language instructions for action planning. The depth supervision could be complementary to low-level instructions because the supervision of HLSM provides rich visual information by depth, which contains visual details, while the supervision of our HACR provides abstract language information for high level guidance for actions. We argue that it is non trivial to claim which supervision is less or more. We add this discussion contrasting HACR with HLSM in the Sec. A.5 in the revision.
>
>
> > **Overfitting issue.** The performance of HACR seems to suffer from strong overfitting in table 1 (from seen to unseen and from validation to test). More analysis for such a performance gap is needed and makes the paper stronger.
>
> $\to$ A possible reason that HACR shows strong performance in seen but less in unseen is insufficiency of data. Due to the limited data size for imitation learning of the given benchmark (note that despite its size being largest in the literature, it is an offline dataset, *i.e.*, the training data are obtained from the simulator by *a single expert trajectory* not using the simulator to learn various possible scenarios) and large language domain gap in seen/unseen environments, generalization is challenging. As the detailed supervision about the environment, *e.g.*, depth information, may help generalization (HLSM), using both supervision for environments and detailed language would reduce the generalization gap further. We appreciate the suggestion for making the paper stronger, and add it in 5th para in Sec. A.5.

---

> > ### Comment · Reviewer_pj1s · 2021-11-25
> > **Reviews Update**
> >
> > The reviewer appreciates the authors' effort to explain the concerns and resolves some of the reviewer's concerns. The reviewer still has some serious concerns unresolved. First, it is still somewhat unreasonable to claim that "the language instructions would contain sufficient information for high-level subgoal planning and the visual input might contain unnecessary visual details that do not help planning." If you see something, it will be at least not worse to to get the subgoal. A more accurate way may be the current way to encode visual information is not smart enough rather than the visual information is useless. Second, the reason of overfitting in HARC is more likely to be the pure NN model is not efficient as the 3D information used in HLSM. If the reason of the overfitting is mainly from the insufficiency of data augmentation. It should be easy for HACR to use simulated visual scenes. And more fumdamentally, it is about the novelty of the proposed module network for action learning, comparing with previous baselines like [Singh et al., ICCV 2021 and Corona et al. 2020].
> >
> > The reviewer thought the authors have clarified some of my questions. However, the reviewer still has some doubts on the remaining issues. The reviewer would like to raise rating from 3 to 4. However, there is no 4, thus the reviewer keeps a rating of 3.

---

> > > ### Author Response · Authors · 2021-11-26
> > > **Responses to the updated review**
> > >
> > > Thank you for sharing your concerns. We reply to the detailed concerns below.
> > >
> > >
> > > > First, it is still somewhat unreasonable to claim that "the language instructions would contain sufficient information for high-level subgoal planning and the visual input might contain unnecessary visual details that do not help planning." If you see something, it will be at least not worse to get the subgoal. A more accurate way may be the current way to encode visual information is not smart enough rather than the visual information is useless.
> > >
> > > $\to$ We agree with your intuition of (1) “if you see something, it will be at least not worse to get the subgoal” *for humans*, and it is always possible that (2) the current way of encoding visual information may be not smart enough and the visual information may be useful for subgoal planning.
> > >
> > > However, for (1), we respectfully argue that the intuition for humans may not hold for a machine learning model and *vice versa*. If the visual information contains unnecessary details, the machine agent may not be able to filter them out unlike humans and it leads to performance drop (‘causal misidentification’ discusses this (de Haan et al., 2019)).
> > >
> > > For (2), we respectfully argue that the empirical advances by the current model could justify its benefit as a stepping stone to build a better model in the future, similar to (Pashevich et al., 2021), (Singh et al., 2021)  and (Suglia et al., 2021).
> > >
> > >
> > > > Second, the reason of overfitting in HACR is more likely to be the pure NN model is not efficient as the 3D information used in HLSM.
> > >
> > > $\to$ While it could be true, we may possibly guess that the overfitting may not be attributed to the pure NN model. One evidence that the pure NN model could be as efficient (*i.e.*, achieving a goal in short time steps, thus, having better PLW success rate) as ones with 3D information is that (Zhang & Chai, 2021) achieves a higher PLW (path length weighted) success rate in test unseen than the recently released HLSM (5.86 *vs.* 5.55), despite having a lower overall SR than HLSM (13.87% *vs.* 20.27%).
> > >
> > > In addition, please note that HLSM exploits perfect egomotion, leading to perfect camera calibration and therefore to accurate 3D reconstruction. But such 3D information could be noisy in the real world and obtaining an accurate estimate may require additional computation (then it could be a less efficient solution compared to NN-based approaches).
> > >
> > >
> > > > > If the reason of the overfitting is mainly from the insufficiency of data augmentation, it should be easy for HACR to use simulated visual scenes.
> > >
> > > $\to$ It is an interesting suggestion (we guess that the term ‘simulated visual scenes’ refers to scenes obtained by running the simulator. Please correct us if we are wrong). But note that the simulated visual scenes have to be *annotated* with language instructions (*i.e.*, goal statements and step-by-step instructions) to be used for training a model. Unfortunately, these human annotations are expensive (Shridhar et al., 2020). Therefore we respectfully argue that the suggested process may not be trivial since it requires expensive data annotations.
> > >
> > >
> > > > And more fundamentally, it is about the novelty of the proposed module network for action learning, comparing with previous baselines like [Singh et al., ICCV 2021 and Corona et al. 2020].
> > >
> > > $\to$ Please note that ‘not using the visual input for PCC’ is one of the technical details for better empirical results, not one of our main contributions (please cordially refer to our contribution summary at the end of the introduction section), and the ‘overfitting’ is commonly observed in many behavior cloning based methods for the same tasks (Shridhar et al., 2020; Singh et al., 2021), not due to the novelty of the proposed method.

---

> > > > ### Comment · Reviewer_pj1s · 2021-11-27
> > > > **Response Updated**
> > > >
> > > > First of all, I would like to thank the authors for their constant effort to address my concerns.
> > > >
> > > > While some of the concerns have been clarified like the supervision signal, the reviewer still has some doubts on the remaining concerns.
> > > >
> > > > **[Overfitting Issues]**. The reviewer argues that the better performance of HACR on the overfitting issues come from the usage of additional simulated scenes. Actually, such data augmentation for additional visual input have also been used in the previous pure NN methods (e.g. Episodic Transformer[ICCV 2021] and  Singh et al., ICCV 2021) for pre-training **without language input**. It is a weak argument to avoid pertaining by claiming "But note that the simulated visual scenes have to be annotated with language instructions (i.e., goal statements and step-by-step instructions) to be used for training a model.". Also, it has been proved in the contemporaneous work (https://openreview.net/forum?id=qI4542Y2s1D) that such 3D visual representation is extremely effective. Of course, there is no need to compare with this new work since it is a contemporaneous submission.
> > > >
> > > > **[Contribution Clarification]**. Beyond this analysis on the experimental section, the reviewer would like to come back to the claimed contributions at the end of the introduction. First, the contribution of the module networks for instruction following seems to be weak as many previous works (e.g. [Singh et al., ICCV 2021 and Corona et al. 2020]) have been using the idea of module networks all the time. The novelty of OEM and LEM is novel but limited since they are some detailed modules designed specifically for AlfRed. Finally, the proposed framework achieves comparable performance to the current STOA method. Such comparable performance is novel enough to serve as the major contribution.
> > > >
> > > > **[Others]** The current estimation is from the reviewer's understanding and if the reviewer is wrong about anything, please directly point out and the reviewer will re-evaluate the changes.

---

> > > > > ### Author Response · Authors · 2021-11-28
> > > > > **Response to the reviewer’s response updated**
> > > > >
> > > > > Thank you for the further clarification on the concerns. We misunderstood some of the questions and respect the reviewer’s opinion.
> > > > >
> > > > > > **[Overfitting Issues].** The reviewer argues that the better performance of HACR on the overfitting issues come from the usage of additional simulated scenes. Actually, such data augmentation for additional visual input have also been used in the previous pure NN methods (e.g. Episodic Transformer[ICCV 2021] and Singh et al., ICCV 2021) for pre-training without language input. It is a weak argument to avoid pertaining by claiming "But note that the simulated visual scenes have to be annotated with language instructions (i.e., goal statements and step-by-step instructions) to be used for training a model."
> > > > >
> > > > > $\to$ We misunderstood your point, our apologies. We agree that the pretraining by the additional simulated scenes would ease the overfitting issue, similar to ‘Episodic Transformer [ICCV 2021]’ and ‘Singh et al., [ICCV 2021],’ and it does not require additional language annotations. Thank you for the suggestion.
> > > > >
> > > > >
> > > > > > **[Overfitting Issues].** Also, it has been proved in the contemporaneous work (https://openreview.net/forum?id=qI4542Y2s1D) that such 3D visual representation is extremely effective. Of course, there is no need to compare with this new work since it is a contemporaneous submission.
> > > > >
> > > > > $\to$ Thank you for pointing out the work.
> > > > >
> > > > >
> > > > > > **[Contribution Clarification].** Beyond this analysis on the experimental section, the reviewer would like to come back to the claimed contributions at the end of the introduction. First, the contribution of the module networks for instruction following seems to be weak as many previous works (e.g. [Singh et al., ICCV 2021 and Corona et al. 2020]) have been using the idea of module networks all the time. The novelty of OEM and LEM is novel but limited since they are some detailed modules designed specifically for AlfRed. Finally, the proposed framework achieves comparable performance to the current STOA method. Such comparable performance is novel enough to serve as the major contribution.
> > > > >
> > > > > $\to$ The reviewer’s understanding is correct and we respect the reviewer’s opinion.

---

### Official Review · Reviewer_wh6P · 2021-11-13

**Correctness:** 4
**Technical Novelty And Significance:** 3
**Empirical Novelty And Significance:** 4
**Recommendation:** 6
**Confidence:** 4

**Main Review:**

Strengths:
- In one of their experiments, they show that training a hierarchical policy learns faster and more efficient action sequences as opposed to their flat policy counterparts. Also, hierarchical policy produces subgoals which leads to a more interpretable and transparent approach.
-	The paper has good results with their method beating prior state-of-the-art methods on mostly all metrics. They also provide exhaustive ablations showing the importance of each component in their approach.
- The paper is well-written and easy to follow with proper figures and with explanations of all the notations used in various modules. This is especially helpful since there are a bunch of moving parts in the approach.

Weaknesses/Clarifications:

- What was the value of $W$ chosen for experiments in eq (3)? I believe it is not mentioned in the paper. How was it decided and how does it impact the usability of Loop Escape module for the ALFRED task?
-	What are the 7 subgoals considered in the Policy Composition Controller module and how were they decided? It is unclear to me of how does the sub-instruction $\hat{x}_i$ correspond to a single subgoal? For the sub-instruction (3) in language instruction in Figure (2) _“Turn around, bring the potato to the microwave on the right.”_, this corresponds to two subgoals “GoToLocation” and “HeatObject”.
-	In figure (2), there are two “Manipulate” action blocks, one in PCC module and another as a predicted navigation action in Master policy. Based on the context, it seems that Manipulate in PCC passes the control to either Master Policy or Interaction Policies depending on whether the subgoal prediction from PCC was an interaction subgoal or not. But “Manipulate” in the Master Policy seem to be doing the same thing. How are they different from each other? It might be good to clarify. Also, it seems that “Manipulate” in Master Policy would act as a “Stop Token” to end the navigation action sequence, does there have to an interaction action after that? What would happen if PCC predicts 2 consecutive “GoToLocation” in a row, but the end of Master Policy triggers an Interaction Policy action due to the “Manipulate” action?


**Summary Of The Paper:**

This paper proposes a hierarchical approach for policy learning in interactive instruction following for the ALFRED task. They infer a series of subgoals to be executed from the language instructions. Then, based on the subgoals predicted, they run Master policy which executes navigation actions, followed by Interaction policies which produce interactions to be done in the environment. They propose Object Encoding Module which provides target object information and serves as navigation subgoal monitor. They also propose a Loop Escape Module which based on the similarities between visual features detects if a loop has been made.

**Summary Of The Review:**

Overall, I feel that this is a good paper with good results and clear presentation. I am happy to increase my score if my concerns can be answered.

---

> ### Author Response · Authors · 2021-11-21
> **Answers to the questions of Reviewer wh6P**
>
> Thank you for the encouraging remarks on an interpretable and transparent idea, good results, exhaustive ablations, and clear presentation. We address your concerns as follows.
>
>
> > **Value of $W$ chosen for experiments in eq (3)?** How was it decided and how does it impact the usability of Loop Escape module for the ALFRED task?
>
> $\to$ We obtain $W=10$ by a grid search (from 0 to 25 with the step of 5) on the validation set. We observe that variation over different values is not large (up to 0.36% fluctuation), implying that the performance is not very sensitive to this hyperparameter. We add this sensitivity analysis in Figure 5 in Sec. A.6 in revision.
>
>
> > **What are the 7 subgoals considered in the Policy Composition Controller module and how were they decided?**
>
> $\to$ The 7 subgoals are GotoLocation, PickupObject, PutObject, CoolObject, HeatObject, CleanObject, SliceObject, and ToggleObject. For our evaluation with ALFRED, we used the set defined in the benchmark.
>
>
> > **How does the sub-instruction $x^i$ correspond to a single subgoal?** For the sub-instruction (3) in language instruction in Figure (2) *“Turn around, bring the potato to the microwave on the right.”*, this corresponds to two subgoals “GoToLocation” and “HeatObject”.
>
> $\to$ For our experiments, we map a single sub-instruction $x^i$ to a single subgoal $s_i$. We found this given example confusing. In the example, the instruction seems to contain two subgoals (‘GoToLocation’ and ‘HeatObject’) but it actually directs one subgoal (‘GoToLocation’ by ‘Turning around and go’ to the microwave with the potato) because it directs the agent to navigate to the microwave with a potato (already possessing it) and does *not direct to heat the potato yet* (though we may imply that it may heat). We change the example sentence in Figure 2 to a less confusing one from the dataset; *’’Turn left and walk to the microwave‘’*.
>
>
> > **Two “Manipulate” blocks in figure (2)**
> > > (a) one in PCC and another in master policy predictions.** From context, it seems that “Manipulate” in PCC passes control to MP or IP based on PCC’s prediction. The “Manipulate” in MP also seems to do the same. How are they different? Clarify.
>
> $\to$ “Manipulate” in the master policy is an *action token* that indicates the *end of navigation* (you’re correct. Please refer to our answer to your next question for details) whereas “Manipulate” in PCC determines whether the subgoal is a manipulation one or not. We found this confusing. So, we replace the notation of “Manipulate” in PCC to ‘’GotoLocation?’’ in Figure 2 in the revision for clarity.
>
>
> > > (b) It seems that “Manipulate” in Master Policy would act as a “Stop Token” to end the navigation action sequence, does there have to be an interaction action after that?**
>
> $\to$ Yes, the “Manipulate” in “Master Policy” acts as a “Stop Token” to terminate navigation and pass the control to the “interaction Policy,” which initiates interaction actions.
>
>
>
> > > (c) What would happen if PCC predicts 2 consecutive “GoToLocation” in a row, but the end of Master Policy triggers an Interaction Policy action due to the “Manipulate” action?
>
> $\to$ We regard this as a failure. Currently, we do not have a way to prevent it. A mechanism to recover from such failure is an interesting future work.

---

### Author Response · Authors · 2021-11-22
**Summary of revision**

**Revisions in the draft:**

Based on the reviewers’ comments, we rearrange some of the sections to add the additional details and to improve readability of the paper. We've highlighted the revised content in ‘Red’ to make it easier to read.

Summary of changes:
- Table 2 $\to$ Tabele 2 and Table 3
- Section 4.4 (a, d, e) $\to$ Section A.4.1 (a,b,c)
- Input Ablations: Section A.4 $\to$ A.4.2
- Additional Experimental Results: Section A.5 $\to$ A.6
- Figure 5-12 $\to$ Figure 7-14
- Table 4 $\to$ Table 5 in appendix
- Table 5 $\to$ Table 6
- An additional variable ‘T’ is introduced to denote the index of the current subgoal. Modifications in Figure 2 and Equations are made accordingly.
- Eq. 6 $\to$ Eq. 7

\
For all reviewers (wh6P, pj1s, KXWq, BjKc, 4rTh),

> Section 1 (paragraph 4)\
$\to$ We add motivation regarding the object encoding module with references to prior works.
[For reviewer BjKc]

> Section 3 (paragraph  1)\
$\to$ We add motivation regarding processing navigation and interaction separately using specialised modules.
[For reviewer BjKc]

>Section 3.1 (paragraph 1)\
$\to$ We modify the section to reduce the claims for interpretability for PCC as suggested.
[For reviewer BjKc]

>Figure 2\
$\to$ We revise Figure (2) with a different name for the *Manipulate* block in PCC (‘’Manipulate’’ $\to$ ‘’GotoLocation?’’) to contrast it from the *Manipulate* action predicted by master policy. [For reviewer wh6P, KXWq]\
$\to$ We add a separate variable *T* to denote the index of current subgoal.
[For reviewer wh6P, KXWq]

>Section 3.2.1\
$\to$ We improved the details about the processing of language instructions for creating the subtask combinations for navigation. We clarify the motivation for the *subtask language combinations*.
[For reviewer KXWq]

> Table 1\
$\to$  We modify Table 1 to include details about different types of supervision used by each of the prior works.
[For reviewer pj1s]

> Section 4.3 (paragraph 1)\
$\to$ We modify the section with better justification.
[For reviewer BjKc]

> Table 2 and Section 4.4\
$\to$ Table 2 is modified to include the ablations about the main contributions (MIP, NIH, OEM) of the work only. Ablation study for the design components (OCMP, FIP and DA) is shifted to table 3 in appendix. [For reviewer KXWq]\
$\to$ The details about the architectural component ablations is also shifted to Appendix in Section A.4.1 [For reviewer KXWq]

> Section A.1\
$\to$ We add details about interaction policy training including the objective functions.
[For reviewer 4rTh]

> Section A.3
>> (Dataset and Metrics)\
$\to$ We add the details about the seven subgoals used for the task.
[For reviewer wh6P]\
>> (Implementation details)\
$\to$ We add the value and selection procedure for the loop escape module’s hyperparameter W.
[For reviewer wh6P, BjKc]

> Section A.4.1 and Table 3\
$\to$ We add the ablation study and empirical results for the design components (OCMP, FIP and DA)  of HACR taken from TTable 2 in the original draft.
[For reviewer KXWq]

> Section A.5 (Additional Related Works)\
$\to$ We add a detailed comparison about technical aspects of prior works employing hierarchical approach (Blukis et al., 2021; Zhang & Chai, 2021).
[For reviewer KXWq]

 > Section A.6 (paragraph 1)\
$\to$ We add a detailed empirical comparison with the prior works employing hierarchical approach (Blukis et al., 2021; Zhang & Chai, 2021).
[For reviewer KXWq]

> Table 5\
$\to$ We modify the table to include results for (Blukis et al., 2021; Zhang & Chai, 2021) for different task types of varying lengths and complexity.
[For reviewer KXWq]

> Section A.6.1 and table 7\
$\to$ We add an ablation study on visual input for the policy composition controller (PCC).
[For reviewer pj1s]

> Section A.6.2 and table 8:\
$\to$ We add an ablation study on visual input for master policy (MP).
[For reviewer pj1s]

> Section A.6.3 and Figure 5\
$\to$ We add a sensitivity analysis for the loop escape module’s (LEM) hyperparameter W.
[For reviewer wh6P, BjKc]

> Section A.7\
$\to$ We add the discussion about *What to learn and what to bake in as an inductive bias?*
[For reviewer BjKc]

> Section A.8\
$\to$ We add a discussion about the hardware, training, and computational cost for more complicated interaction policies or a large number of subgoals.
[For reviewer 4rTh]

> Figure 6\
$\to$ Learning curve for the used subgoal policies.
[For reviewer 4rTh]



\
**Reproducibility Statement**
We take the reproducibility of research very seriously and describe the algorithm in detail in Sec. 3 and Sec. A.1. Please find the code in the supplementary material.
We also solemnly promise to release all codes,  environment information,  learned models, and complete task configurations in a public repository.

---

### Decision · Program_Chairs · 2022-01-20

**Decision:**

Reject

**Comment:**

This paper led to significant discussion, and the AC is generally on the fence. First of all, thanks to the reviewers for the significant time they invested in the discussion, and thanks for the authors for promptly and patiently answering our questions.

Overall, the reviewer recommendations are positive. However, the discussion showed that despite the positive recommendation, the reviewers struggled to distill the general contribution of the paper beyond performance on ALFRED. In discussion, the authors distinguished their contribution from existing work by focusing on using a set of low-level policies at the root of the overall policy. This relies on the discrete set of behaviors that is defined within the ALFRED benchmark. It's not clear how it generalizes to the actual problem of instructing a robot to execute natural language instruction. In realistic scenarios, is it possible to define a set of behaviors in such a clean way, and at scale? And then train/manage a separate model for each behavior? The set of interaction policies in Figure 2 illustrates this challenge well. The answer to this scaling question is not clear. This corresponds to a concern raised repeatedly by the reviewers about the approach too specialized to ALFRED. The AC shares this concern.

(which are roughly equal to the SOTA at the time of submission, but show significantly more overfitting to seen environments)

On the positive side, this is solid work, with good results. The paper is well written, and the authors largely addressed the concerns raised as much as possible. The results are not SOTA though. The current SOTA was submitted on 09/19/2021, prior to the ICLR deadline -- it's not included in the results table in this paper. (To clarify, the fact that it's not the current SOTA does not affect the final decision, as they are considered as contemporaneous.) With concerns regarding the specificity of the approach, this paper may interest researchers working on ALFRED, but not clear to what depth, despite the clearly significant work and effort the authors put into the paper.

(If the paper is accepted, the AC asks the authors to fix the standing errors with regard to previous work, as discussed below, and to include more recent results from the leaderboard)